# Free Probability for predicting the performance of feed-forward fully connected neural networks

**Reda CHHAIBI**[1]    **Tariq DAOUDA**[2]    **Ezéchiel KAHN**[3]

[1] Institut de Mathématiques de Toulouse, Université Paul Sabatier, Toulouse, France.
[2] Institute of Biological Sciences, Mohammed VI Polytechnic University, Benguerir, Morocco.
[3] CERMICS, Ecole des Ponts, INRIA, Marne-la-Vallée, France.

reda.chhaibi@math.univ-toulouse.fr,
tariq.daouda@um6p.ma, ezechiel.kahn@enpc.fr

## Abstract

Gradient descent during the learning process of a neural network can be subject to many instabilities. The spectral density of the Jacobian is a key component for analyzing stability. Following the works of Pennington et al., such Jacobians are modeled using free multiplicative convolutions from Free Probability Theory (FPT). We present a reliable and very fast method for computing the associated spectral densities, for given architecture and initialization. This method has a controlled and proven convergence. Our technique is based on an homotopy method: it is an adaptative Newton-Raphson scheme which chains basins of attraction. In order to demonstrate the relevance of our method we show that the relevant FPT metrics computed before training are highly correlated to final test accuracies – up to 85%. We also nuance the idea that learning happens at the edge of chaos by giving evidence that a very desirable feature for neural networks is the hyperbolicity of their Jacobian at initialization.

## 1  Introduction

Neural network training and tuning can be wasteful in human and energy resources. For example [23, Table 1] show that a single GPU training can have high energetic costs. Often, this is nothing compared to architecture search. In this context, an obvious moonshot is estimating the performance of an architecture *before* training. The goal of this paper is more realistic: providing a fast and reliable computational method for estimating stability before training – and stability is a good proxy for performance [12].

**Framework:** Consider a feed-forward network of depth $L \in \mathbb{N}$, with $L$ full-connected layers. For each depth $\ell \in \{1, 2, \ldots, L\}$, the layer has activation vector $x_\ell \in \mathbb{R}^{N_\ell}$, where $N_\ell$ is the current width. The vector $x_0 \in \mathbb{R}^{N_0}$ takes in the neural network's input, while $x_L \in \mathbb{R}^{N_L}$ gives the output. The vector of widths is written $\mathbf{N} := (N_0, N_1, \ldots, N_L)$ and will appear in superscript to indicate the dependence in any of the $N_\ell$'s. The following recurrence relation holds between layers $x_\ell = \phi_\ell \left( W_\ell^{(\mathbf{N})} x_{\ell-1} + b_\ell^{(\mathbf{N})} \right)$, where $\phi_\ell$ is a choice of non-linearity applied entry-wise, $W_\ell^{(\mathbf{N})} \in M_{N_\ell, N_{\ell-1}}(\mathbb{R})$ is a weight matrix and $b_\ell^{(\mathbf{N})} \in \mathbb{R}^{N_\ell}$ is the vector of biases. We write $h_\ell := W_\ell^{(\mathbf{N})} x_{\ell-1} + b_\ell^{(\mathbf{N})}$ for the pre-activations.

The Jacobian computed during back-propagation can be written explicitly by using the chain rule:

$$J^{(\mathbf{N})} := \frac{\partial x_L}{\partial x_0} = \frac{\partial x_L}{\partial x_{L-1}} \frac{\partial x_{L-1}}{\partial x_{L-2}} \cdots \frac{\partial x_1}{\partial x_0} = D_L^{(\mathbf{N})} W_L^{(\mathbf{N})} D_{L-1}^{(\mathbf{N})} W_{L-1}^{(\mathbf{N})} \ldots D_1^{(\mathbf{N})} W_1^{(\mathbf{N})} , \quad (1.1)$$

36th Conference on Neural Information Processing Systems (NeurIPS 2022).

where $D_\ell$'s are the diagonal matrices given by

$$\left[D_\ell^{(\mathbf{N})}\right]_{i,i} = \phi'_\ell([h_\ell]_i) . \tag{1.2}$$

Technically, a step of gradient descent updates weights and biases following

$$\left(W_\ell^{(\mathbf{N})}, b_\ell^{(\mathbf{N})}\right) \leftarrow \left(W_\ell^{(\mathbf{N})}, b_\ell^{(\mathbf{N})}\right) - \alpha \frac{\partial \mathcal{L}}{\partial(W_\ell^{(\mathbf{N})}, b_\ell^{(\mathbf{N})})} , \tag{1.3}$$

for each $\ell = 1, \ldots, L$. Here $\alpha > 0$ is the learning rate and $\mathcal{L}$ is the loss on a minibatch. If the minibatch has size $B \in \mathbb{N}$, and corresponds a small sample $((X_i, Y_i) ; i = 1, \ldots, B)$ of the dataset, we have $\mathcal{L} = \frac{1}{B} \sum_{i=1}^{B} d(x_L(X_i), Y_i)$ . Here $d$ is a real-valued distance or similarity function, the $X_i$'s are the input vectors while the $Y_i$'s are the output vectors (e.g. labels in the case of classifier, $Y_i \approx X_i$ in the case of an autoencoder etc...).

The chain rule dictates:

$$\frac{\partial \mathcal{L}}{\partial(W_\ell^{(\mathbf{N})}, b_\ell^{(\mathbf{N})})} = \frac{\partial \mathcal{L}}{\partial x_L} \frac{\partial x_L}{\partial x_{L-1}} \cdots \frac{\partial x_{\ell+1}}{\partial x_\ell} \frac{\partial x_\ell}{\partial(W_\ell^{(\mathbf{N})}, b_\ell^{(\mathbf{N})})} = \frac{\partial \mathcal{L}}{\partial x_L} J_\ell^{(\mathbf{N})} \frac{\partial h_\ell}{\partial(W_\ell^{(\mathbf{N})}, b_\ell^{(\mathbf{N})})} , \tag{1.4}$$

where

$$\frac{\partial \mathcal{L}}{\partial x_L} = \frac{1}{B} \sum_{i=1}^{B} \partial_1 d(x_L(X_i), Y_i) \in M_{1, N_L}(\mathbb{R}) , \tag{1.5}$$

$$J_\ell^{(\mathbf{N})} = D_L^{(\mathbf{N})} W_L^{(\mathbf{N})} \cdots D_{\ell+1}^{(\mathbf{N})} W_{\ell+1}^{(\mathbf{N})} D_\ell^{(\mathbf{N})} \in M_{N_L, N_\ell}(\mathbb{R}) . \tag{1.6}$$

Therefore, for the sake of simplicity, we shall focus on the Jacobian $J^{(\mathbf{N})}$ given in Eq. (1.1) since it has exactly the same form as the $J_\ell^{(\mathbf{N})}$ given in Eq. (1.6). The issue is that a large product of (even larger) matrices can easily become unstable. If many singular values are $\ll 1$, we have gradient vanishing. If many singular values are $\gg 1$, we have gradient explosion. Such a transition can be referred to as the edge of chaos [26, 27].

**Intuition.** This instability is easily understood thanks to the naive analogy with the one-dimensional case. Indeed, the geometric progression $q^n$ with $n \to \infty$ is the archetype of a long product and it converges extremely fast, to either 0 if $|q| < 0$ or to $\infty$ if $|q| > 1$.

A less naive intuition consists in observing that mini-batch sampling in Eq. (1.5) is very noisy. It is fair to assume that $\frac{\partial \mathcal{L}}{\partial x_L}$ has a Gaussian behavior with covariance proportional to $I_{N_L}$ – either because of the Central Limit Theorem if $B$ is large enough or after time averaging, because of the mixing properties of SGD [6, 8]. Therefore, each gradient step $\alpha \frac{\partial \mathcal{L}}{\partial(W_\ell^{(\mathbf{N})}, b_\ell^{(\mathbf{N})})}$ in Eq. (1.3) is approximately a Gaussian vector with covariance proportional to:

$$\alpha^2 \left(\frac{\partial h_\ell}{\partial(W_\ell^{(\mathbf{N})}, b_\ell^{(\mathbf{N})})}\right)^T \left(J_\ell^{(\mathbf{N})}\right)^T J_\ell^{(\mathbf{N})} \frac{\partial h_\ell}{\partial(W_\ell^{(\mathbf{N})}, b_\ell^{(\mathbf{N})})} .$$

Simplifying further, we see the importance of the spectrum of $\left(J_\ell^{(\mathbf{N})}\right)^T J_\ell^{(\mathbf{N})}$ for stability. Basically, eigenvectors of $\left(J_\ell^{(\mathbf{N})}\right)^T J_\ell^{(\mathbf{N})}$ are the directions along which the one-dimensional intuition applies.

**Randomness.** Starting from the pioneering works of Glorot and Bengio [9] on random initializations, it was suggested that the spectral properties of $J^{(\mathbf{N})}$ are an excellent indicator for stability and learning performance. In particular, an appropriate random initialization was suggested and since implemented in all modern ML frameworks [19, 1].

We make classical choices of random initializations. The biases $b_\ell^{(\mathbf{N})}$ are taken as random vectors which entries are centered i.i.d. Gaussian random variables with standard deviation $\sigma_{b_\ell}$. For the

weights, we will consider the following matrix ensembles: the $[W_\ell^{(\mathbf{N})}]_{i,j}$ are drawn from i.i.d. centered random variables with variance $\sigma^2_{W_\ell}/N_\ell$ and finite fourth moment as in [17].

**Modeling spectrum thanks to Free Probability Theory.** Now, following the works of Pennington et al. [20], the tools of Free Probability Theory (FPT) can be used to quantitatively analyze the singular values of $J^{(\mathbf{N})}$ in the large width limit. The large width limit is particularly attractive when studying large deep networks, especially because free probability appears at relatively small sizes because of strong concentration properties [14]. Indeed, random matrices of size 100 exhibit freeness.

For the purposes of this paragraph, we restrict ourselves to square matrices and assume $N_\ell = N$ for all $\ell = 1, \ldots, L$. In fact, FPT is concerned with the behavior of spectral measures as $N \to \infty$. For any diagonalizable $A_N \in M_N(\mathbb{R})$, the associated spectral measure on the real line is:

$$\mu_{A^{(N)}}(dx) := \frac{1}{N} \sum_{i=1}^{N} \delta_{a_i^{(N)}}(dx)$$

with the $a_i^{(N)}$'s being the eigenvalues of $A_N$. For ease of notation, the spectrum of (squared) singular values is written $\nu_{A^{(N)}} := \mu_{\left(A^{(N)}\right)^T A^{(N)}}$. A fundamental assumption for invoking tools from Free Probability Theory, is the assumption of *asymptotic freeness*. Without defining the notion, which can be found in [16], let us describe the important computation it allows, discovered in the seminal work of Voiculescu [25]. Given two sequences of square matrices $A^{(N)}, B^{(N)}$ in $M_N(\mathbb{R})$, with converging spectral measures:

$$\lim_{N \to \infty} \nu_{A^{(N)}} = \nu_A , \quad \lim_{N \to \infty} \nu_{B^{(N)}} = \nu_B ,$$

we have that, under the assumption of asymptotic freeness $\lim_{N \to \infty} \nu_{A^{(N)} B^{(N)}} = \nu_A \boxtimes \nu_B$, where $\boxtimes$ is a deterministic operation between measures called multiplicative free convolution. The $\boxtimes$ will be detailed in Section 2. The letter $A$ (as well as $B$) does not correspond to a limiting matrix but to an abstract operator, with associated spectral measure $\mu_A$ and measure of squared singular values $\nu_A$. For such limiting operators, we drop the superscript $(N)$.

Under suitable assumptions which are motivated and detailed later following the works of [20, 11, 18, 17, 7], for all $\ell = 1, \ldots, L$, the measures $\nu_{W_\ell^{(\mathbf{N})}}$ and $\nu_{D_\ell^{(\mathbf{N})}}$ will respectively converge to $\nu_{W_\ell}$ and $\nu_{D_\ell}$. Again the $W_\ell$'s and $D_\ell$'s are abstract operators which only make sense in the infinite width regime. In the limit, asymptotic freeness will also hold. Therefore, we will see that the measure of interest is:

$$\lim_{N \to \infty} \nu_{J^{(\mathbf{N})}} = \nu_J := \nu_{D_L} \boxtimes \nu_{W_L} \boxtimes \cdots \boxtimes \nu_{D_1} \boxtimes \nu_{W_1} . \tag{1.7}$$

The goals of this paper are (1) To give a very fast and stable computation of $\nu_J$, in the more general setup of rectangular matrices (2) Empirically demonstrate that FTP metrics computed from $\nu_J$ do correlate to the final test accuracy.

## 1.1 Contributions

We aim at streamlining the approach of Pennington et al. by providing the tools for a systematic use of FPT. The contributions of this paper can be categorized as:

- Theoretical: In Pennington et al., a constant width is assumed. We generalize the model to allow for varying width profiles, which is more inline with design practices. This requires us to develop a rectangular multiplicative free convolution.

  Then we propose a computational scheme for computing spectral densities, named "Newton lilypads". The method relies on adaptive inversions of $S$-transforms using the Newton-Raphson algorithm. If the Newton-Raphson scheme is only local, we achieve a global resolution by chaining basins of attractions, thanks to doubling strategies. As such, we have theoretical guarantees for the convergence.

  Interestingly, even in the FPT community, inverting $S$-transforms has been considered impossible to realize in practice. As we shall see, an $S$-transform is a holomorphic map with multi-valued inverse. In the words of [5, p.218], "the operations which are necessary in this method (the inversion of certain functions, the extension of certain analytic functions) are almost always impossible to realise practically."

- Numerical: This misconception led to the use of combinatorial methods based on moments, or fixed-point algorithms via the subordination method [2, 15, 24]. In the ML community, Pennington et al. pioneered the application of FPT to the theoretical foundations of Machine Learning and did not shy away from inverting $S$-transforms. Their [20, Algorithm 1] is based on a generic root finding procedure, and choosing the root closest to the one found for the problem with one less layer. A major drawback of this method is that there is no guarantee to find the correct value, unlike our chaining which always chooses the correct branch.

  Not only Newton lilypads has theoretical guarantees of convergence, but it is also an order of magnitude faster (Fig. 1.1). A few standard Cython optimizations allow to gain another order of magnitude, although this can certainly be refined.

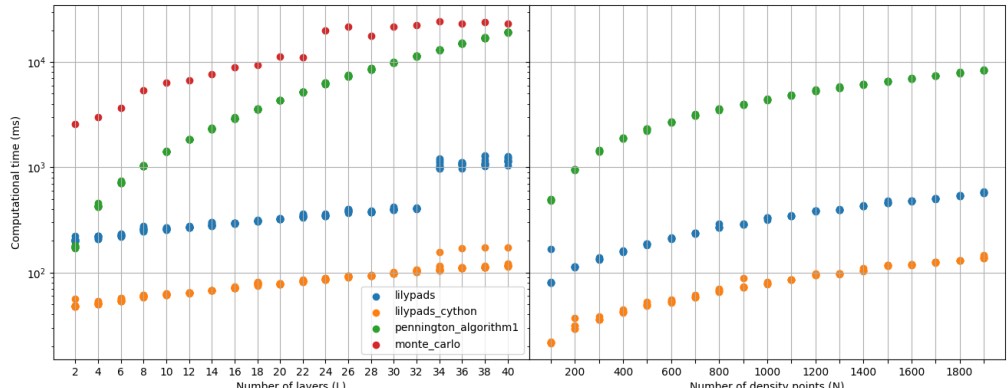

Figure 1.1: Computation time (in ms) for the density of $\nu_J$ w.r.t. depth $L$ (left) and number of density points (right). Vertical axis is log-scale. The benchmarked methods are Newton lilypads in pure Python (blue), Newton lilypads with Cython optimizations (orange), Pennington et al.'s Algorithm 1 using a native root solver (green), Monte-Carlo (red). While of a different nature, Monte-Carlo is given for indication.

- Empirical: We analyze the correlation between the test accuracy of several randomly generated MLP architectures and the quantiles of $\nu_J$, after a fixed number of epochs. The same architectures were independently trained on the MNIST, FashionMNIST and CIFAR10 datasets. We find that accuracy is strongly correlated to FTP quantiles of $\nu_J$ (see Table 1.1). Remarkably, the correlation is almost entirely captured by the higher quantiles – see Table 1.2 for individual $R$ factors. Scatter plots showing the distribution of test accuracy, $log_{10}$ of the 90th quantile and number of parameters can be seen in Fig. 1.2. Interestingly, smaller networks can perform better than larger one provided they have more spread-out $\nu_J$ distribution. This suggests that spread-out spectral distributions $\nu_J$ are more desirable, provided of course we avoid the vanishing and explosive regimes. In the language of dynamical systems, we say that the Jacobian needs to be hyperbolic i.e. with both contracting and expanding directions. This considerably nuances the idea that learning happens at the edge of chaos. A similar point was made in the conclusion of [6], using the language of hyperbolic attractors.

For reproducibility of the numerical and empirical aspects, a complete implementation is provided in a Github repository

`https://github.com/redachhaibi/FreeNN`

## 1.2 Structure of the paper

We start in Section 2 by stating facts from Free Probability Theory. Most of it is available in the literature, except the definition of the product of rectangular free matrices. To the best of our knowledge, this is novel. There, we establish in the rectangular setting an analogue of Eq. (1.7) in Theorem 2.3.

| Regression factors | Corr. indicator \ Dataset | MNIST | FashionMNIST | CIFAR10 |
|---|---|---|---|---|
| 90th quantile only | Spearman (p-value) | 0.83 (3e-49) | 0.79 (6e-45) | 0.73 (4e-18) |
|  | Pearson (p-value) | 0.84 (9e-54) | 0.81 (1e-48) | 0.74 (3e-19) |
|  | $R$ factor | 0.84 | 0.81 | 0.75 |
| All quantiles of $\nu_J$ | $R$ factor | 0.89 | 0.87 | 0.83 |

Table 1.1: Correlation between test accuracy and FPT metrics. 200 randomly generated MLP architectures were trained on 3 datasets. We performed linear regressions of test accuracy against quantiles of $\nu_J$. Although correlation is stronger considering all quantiles, higher quantiles are individually the most correlated (see Table 1.2). Hence, the computation of Spearman, Pearson and $R$ correlation factors between accuracy and 90th quantile. The last row shows the correlation of accuracy against all the quantiles $10, 20, \ldots, 80, 90\%$ during a multivariate regression.

| Quantile of $\nu_J$ | MNIST | FashionMNIST | CIFAR10 |
|---|---|---|---|
| 10% | 0.058 | 0.035 | -0.016 |
| 20% | 0.084 | 0.055 | 0.005 |
| 30% | 0.176 | 0.134 | 0.063 |
| 40% | 0.204 | 0.156 | 0.089 |
| 50% | 0.234 | 0.186 | 0.123 |
| 60% | 0.301 | 0.244 | 0.196 |
| 70% | 0.407 | 0.342 | 0.304 |
| 80% | 0.597 | 0.539 | 0.498 |
| 90% | 0.845 | 0.807 | 0.755 |

Table 1.2: Accuracy vs ($log_{10}$ of) a single quantile of $\nu_J$. For each dataset and each quantile, the table reports the R factor in a bivariate linear regression.

In Section 3, we explain in detail the FPT model for random neural networks. Then thanks to the results of [18, 17] and our rectangular setting, we show that the spectral measure of the Jacobian $J^{(\mathbf{N})}$ converges to $\nu_J$ and we encode the limit in explicit generating series in Theorem 3.1. This gives how $\nu_J$ can *theoretically* be recovered.

Section 4 presents the numerical resolution which inverts the (multi-valued) generating series. By chaining different (local) basins of attractions, we obtain a global resolution method. Our algorithm is detailed in Algorithm 1 and Theorem 4.1 states the theoretical guarantees.

Finally Section 5 presents the experiment leading to Table 1.1. More details are given in Appendix F, including comments on the benchmark of Fig. 1.1.

## 2 Free Probability

### 2.1 Definitions and notations

Free Probability Theory provides a framework to analyze eigenvalues and singular values of large random matrices. We now introduce various complex-analytic generating series which encode the measures and the basic operations on them. First, the Cauchy-Stieltjes transform of $\mu$, a probability measure on $\mathbb{R}_+$ is:

$$
\begin{aligned}
G_\mu : \quad \mathbb{C}_+ &\rightarrow \quad \mathbb{C}_- \\
z &\mapsto \quad \int_{\mathbb{R}_+} \frac{\mu(dv)}{z-v} \ ,
\end{aligned}
$$

where $\mathbb{C}_\pm := \{z \in \mathbb{C} \mid \pm \Im z > 0\}$ . The transform $G_\mu$ encodes the measure $\mu$ and reciprocally, the measure can be recovered thanks to:

**Lemma 2.1** (Cauchy-Stieltjes inversion formula – Theorem 6 in [16])**.** *We have the weak convergence of probability measures:*

$$
\lim_{y \to 0} -\frac{1}{\pi} \Im G_\mu(x + iy) dx = \mu(dx) \ .
$$

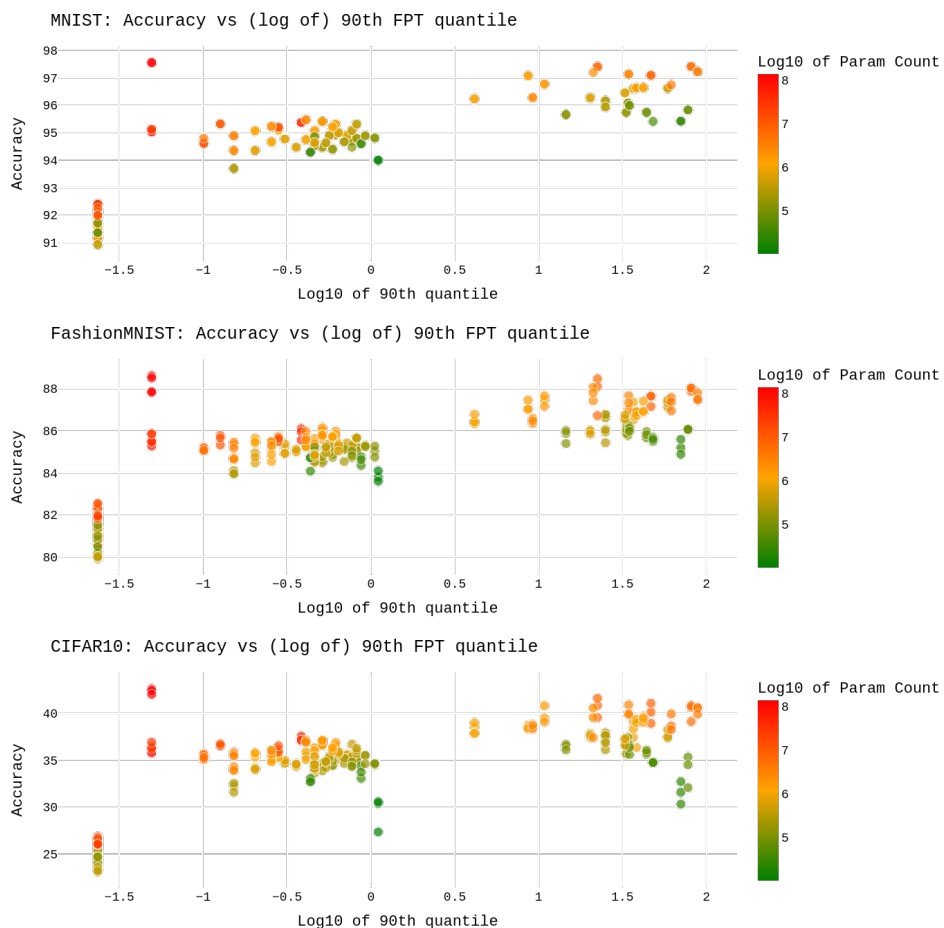

Figure 1.2: Accuracy versus ($log_{10}$ of) 90th quantile for various datasets. From top to bottom, datasets are MNIST, FahsionMNIST, CIFAR10.

The moment generating function is

$$M_\mu(z) = z G_\mu(z) - 1 = \sum_{k=1}^{+\infty} \frac{m_k(\mu)}{z^k} \, , \tag{2.1}$$

where for all $k \in \mathbb{N}$, $m_k(\mu) := \int_\mathbb{R} x^k \mu(dx)$ is the $k$-th moment of $\mu$. For $\mu \neq \delta_0$, $M_\mu$ is invertible in the neighborhood of $\infty$ and the inverse is denoted by $M_\mu^{\langle -1 \rangle}$. The S-transform of $\mu$ is defined as $S_\mu(m) = \frac{1+m}{m M_\mu^{\langle -1 \rangle}(m)}$, and is analytic in a neighborhood of $m = 0$. Furthermore, the variable $z$ will always denote an element of $\mathbb{C}_+$, while the variables $g$ and $m$ will denote elements in the image of $G_\mu$ and $M_\mu$. For a diagonalizable matrix $A^{(N)} \in M_N(\mathbb{R})$, we write $S_{A^{(N)}} := S_{\mu_{A^{(N)}}}$, $G_{A^{(N)}} := G_{\mu_{A^{(N)}}}$, $M_{A^{(N)}} := M_{\mu_{A^{(N)}}}$.

A landmark result in the field introduces free multiplicative convolution in a natural way, and shows that this operation is linearized by the $S$-transform:

**Theorem 2.2** (Voiculescu, [25]). *Consider two sequences of positive matrices, each element in $M_N(\mathbb{R})$*

$$\left( A^{(N)} \, ; \, N \geq 1 \right) \, , \quad \left( B^{(N)} \, ; \, N \geq 1 \right) \, ,$$

*such that:*

$$\lim_{N \to \infty} \mu_{A^{(N)}} = \mu_A \, , \quad \lim_{N \to \infty} \mu_{B^{(N)}} = \mu_B \, .$$

*Under the assumption of asymptotic freeness for $A^{(N)}$ and $B^{(N)}$, there exists a deterministic probability measure $\mu_A \boxtimes \mu_B$ such that $\lim_{N\to\infty} \mu_{A^{(N)}B^{(N)}} = \mu_A \boxtimes \mu_B$. The operation $\boxtimes$ is the multiplicative free convolution. Moreover*

$$S_{AB}(m) = S_A(m)S_B(m) . \tag{2.2}$$

This convergence akin to a law of large numbers is the key ingredient which allows to build the deterministic model for the back-propagation of gradients in Eq. (1.7).

### 2.2 Product of rectangular free matrices

As a generalization of Eq. (2.2) to rectangular matrices, we state:

**Theorem 2.3.** *Let $(p_N)_{N\geq 1}, (q_N)_{N\geq 1}, (r_N)_{N\geq 1}$, be three sequences of integers satisfying*

$$p_N, q_N, r_N \xrightarrow[N\to\infty]{} \infty, \qquad \frac{r_N}{q_N} \xrightarrow[N\to\infty]{} c > 0 .$$

*Consider for all $N \geq 1$ let $A^{(N)}, B^{(N)}$ be random matrices of respective sizes $p_N \times q_N$ and $q_N \times r_N$ such that the (squared) singular laws of $A^{(N)}, B^{(N)}$ converge weakly. Assuming that $B^{(N)}\left(B^{(N)}\right)^T$ and $\left(A^{(N)}\right)^T A^{(N)}$ are asymptotically free, we have that in the limit $N \to \infty$:*

$$S_{(AB)^T AB}(m) = S_{A^T A}(cm) S_{B^T B}(m) .$$

*Proof.* See the appendix, Subsection C.1 □

Implicitly this defines a rectangular multiplicative free convolution, which could be denoted $\boxtimes_c$ in the spirit of the rectangular free additive convolution [4]. But, in the current setting, this is not a good idea. Indeed, if one defines $\mu_1 \boxtimes_c \mu_2$ as the measure whose $S$-transform is $S_{\mu_1}(c \cdot)S_{\mu_2}$, then a quick computation shows that $\boxtimes_c$ is not associative, i.e., for a triplet $(\mu_1, \mu_2, \mu_3)$ of probability measures and a pair $(c_1, c_2) \in \mathbb{R}_+^* \times \mathbb{R}_+^*$, we generically have:

$$\mu_1 \boxtimes_{c_1} (\mu_2 \boxtimes_{c_2} \mu_3) \neq (\mu_1 \boxtimes_{c_1} \mu_2) \boxtimes_{c_2} \mu_3 .$$

A better idea is to treat the dimension ratio $c$ as part of the data via a central extension:

**Definition 2.4.** *On the set of pairs $(\mu, c)$ such that $\mu$ is a probability measure on $\mathbb{R}_+$ and $c \in \mathbb{R}_+^*$, define the operation $\boxtimes$ as:*
$$(\mu_1, c_1) \boxtimes (\mu_2, c_2) := (\nu, c_1 c_2) ,$$
*where $\nu$ is the unique probability measure such that $S_\nu = S_{\mu_1}(c_2 \cdot)S_{\mu_2}$. This extends the classical definition as the usual free convolution is recovered with $(\mu_1, 1) \boxtimes (\mu_2, 1) := (\mu_1 \boxtimes \mu_2, 1)$.*

Such an operation is associative and will allow a neat formulation of the measure of interest in the upcoming Theorem 3.1, entirely analogous to Eq. (1.7).

## 3 Theoretical resolution of the model

**Width profile:** Pennington et al. [20] consider $N_\ell = N$ for $\ell = 1, 2, \ldots, N$. Here, we consider that the width of layers is not constant across layers, which is mostly the case in practice. Indeed, modern architectures typically have very sophisticated topologies with layers varying in widths.

Let us assume that we are in the infinite width regime in the sense that $N_\ell \to \infty$, for all $\ell = 0, 1, 2 \ldots, L$ with: $\frac{N_{\ell-1}}{N_\ell} \xrightarrow[\mathbf{N}\to\infty]{} \lambda_\ell > 0$. And let us denote $\Lambda_\ell := \lim_{\mathbf{N}\to\infty} \frac{N_0}{N_\ell} = \prod_{k=1}^{\ell} \lambda_k$ , with the convention $\Lambda_0 = 1$.

**FPT limits:** Let $\mathcal{N}$ be a standard Gaussian random variable on the real line $\mathbb{P}(\mathcal{N} \in dx) = \frac{e^{-\frac{x^2}{2}}}{\sqrt{2\pi}} dx$ .

Here $D_\ell^{(\mathbf{N})}$ is diagonal with entries $\phi_\ell'([h_\ell]_i)$ (see Eq. (1.2)), and the pre-activations

$$h_\ell = W_\ell^{(\mathbf{N})} x_{\ell-1} + b_\ell^{(\mathbf{N})} = W_\ell^{(\mathbf{N})} \phi_\ell(h_{\ell-1}) + b_\ell^{(\mathbf{N})}$$

clearly depend on the previous layers. Because of this lack of independence, the standard results of FPT cannot be applied directly i.e. asymptotic freeness does not obviously hold. This is an important subtlety that is addressed in the upcoming Theorem 3.1. Based on an information propagation argument, the papers [21, 22] argue that the entries of $h_\ell$ behave as the i.i.d. samples of a Gaussian distribution with zero mean and variance $q^\ell$. A basic law of large numbers applied to Eq. (1.2) gives a limit for the empirical measure $\mu_{D_\ell} = \lim_{N_\ell \to \infty} \mu_{D_\ell^{(\mathbf{N})}} = \phi'_\ell\left(\sqrt{q^\ell}\mathcal{N}\right)$. Also the recurrence for the variance is:

$$q^\ell = f_\ell\left(q^{\ell-1}\right) = \sigma_{W_\ell}^2 \mathbb{E}\left[\phi_\ell\left(\sqrt{q^{\ell-1}}\mathcal{N}\right)^2\right] + \sigma_{b_\ell}^2, \tag{3.1}$$

with initial condition $q^1 = \frac{\sigma_{W_1}^2}{N_1}\sum_{i=1}^{N_1}(x_0^i)^2 + \sigma_{b_1}^2$.

Recently Pastur et al. completed this heuristic thanks to a swapping trick – see [18, Lemma 3.3] and [17, Remark 3.4]. They proved that, regarding the asymptotical spectral properties of $J^{(\mathbf{N})}$, one can replace each $D_\ell^{(\mathbf{N})}$ by a diagonal matrix with independent Gaussian entries $\sqrt{q^\ell}\mathcal{N}$ independent from the rest. In that setting, one can apply the results on products of asymptotically free matrices which were given in Section 2.

**Theorem 3.1.** *In terms of the rectangular multiplicative free convolution, the measure of (squared) singular values of $J^{(\mathbf{N})}$ converges to*

$$\nu_J = (\nu_{D_L}, 1) \boxtimes (\nu_{W_L}, \lambda_L) \boxtimes \cdots \boxtimes (\nu_{D_1}, 1) \boxtimes (\nu_{W_1}, \lambda_1) . \tag{3.2}$$

*Moreover, the S-transform of $J^T J$ in the infinite width regime verifies*

$$S_{J^T J}(m) = \prod_{\ell=1}^{L}\left[S_{D_\ell^2}\left(\Lambda_\ell m\right) S_{W_\ell^T W_\ell}\left(\Lambda_{\ell-1}m\right)\right] . \tag{3.3}$$

*In particular, under the assumption that the entries of $W_\ell$ are i.i.d. :*

$$S_{J^T J}(m) = \prod_{\ell=1}^{L}\left(S_{D_\ell^2}\left(\Lambda_\ell m\right)\frac{1}{\sigma_{W_\ell}^2}\frac{1}{1+\Lambda_\ell m}\right) , \quad M_{J^T J}^{\langle -1\rangle}(m) = \frac{m+1}{m}\prod_{\ell=1}^{L}\frac{\sigma_{W_\ell}^2(1+\Lambda_\ell m)}{S_{D_\ell^2}\left(\Lambda_\ell m\right)} .$$

*Proof.* See the appendix, Subsection C.2. $\qquad\square$

**Master equation:** In the end, we only need to fix width ratios and non-linearities to form $M_{J^T J}^{\langle -1\rangle}(m)$, and get the master equation which we solve numerically thanks to an adaptive Newton-Raphson scheme. The non-linearities ReLU, Hard Tanh and Hard Sine yield explicit formulas, which can be found in Table B.1 of the appendix. If $W_\ell$ has i.i.d. entries, one finds the explicit master equation:

$$M_{J^T J}^{\langle -1\rangle}(m) = \frac{m+1}{m}\prod_{\ell=1}^{L}\sigma_{W_\ell}^2\left(c_\ell + \Lambda_\ell m\right) , \tag{3.4}$$

where $c_\ell = \frac{1}{2}$ when $\phi_\ell$ is ReLU, $c_\ell = C_\ell = \mathbb{P}\left(0 \leq \mathcal{N} \leq \frac{1}{\sqrt{q^\ell}}\right)$ if $\phi_\ell$ is Hard Tanh and $c_\ell = 1$ if $\phi_\ell$ is Hard Sine.

## 4   Numerical resolution

Here we describe the numerical scheme aimed at computing the spectral density of $J^T J$ in Eq. (1.1). We use the following steps to compute the spectral density at a fixed $x \in \mathbb{R}_+$:

- Because of the Cauchy-Stieltjes inversion formula given in Lemma 2.1, pick a small $y > 0$ in order to compute: $-\frac{1}{\pi}\Im G_{J^T J}(z = x + iy)$. The smaller the better, and in practice our method works for up to $y = 10^{-9}$. Figure A.1 shows the same target distribution but convolved with various Cauchy distributions $y\mathcal{C}$ where $y \in \{1, 10^{-1}, 10^{-4}\}$. This corresponds to computing the density $\frac{-1}{\pi}\Im G_\mu\left(\cdot + iy\right)$ for different $y$'s.

- Because of Eq. (2.1), we equivalently need to compute $M_{J^T J}(z)$.

- $M_{J^T J}^{\langle -1 \rangle}(m)$ is available thanks to the master equation in Theorem 3.1. Therefore, we need to invert $m \mapsto M_{J^T J}^{\langle -1 \rangle}(m)$. This step is the crucial part: $M_{J^T J}^{\langle -1 \rangle}$ is multi-valued and one needs to choose the correct branch.

---

**Algorithm 1** Newton lilypads, chaining basins of attraction

---

**Name:** NEWTON_LILYPADS
**Input:** Image value: $z_{objective} \in \mathbb{C}_+$,    (Optional) Proxy: $(z_0, m_0) \in \mathbb{C}_+ \times \mathbb{C}$.
**Output:** $M(z_{objective})$
\# Find a proxy $(z_0, m_0 = 0)$ using a doubling strategy, if None given
**if** $(z_0, m_0)$ is None **then**
    $m \leftarrow 0$
    $z \leftarrow z_{objective}$
    **while** not IS_IN_BASIN(z, m) **do**
        $z \leftarrow z + i\Im(z) = \Re(z) + i2\Im(z)$ \# Double imaginary part
    **end while**
    $m \leftarrow$ NEWTON_RAPHSON$(z, \text{Guess} = m)$
**else**
    $(z, m) \leftarrow (z_0, m_0)$
**end if**
\# Starts heading towards $z_{objective}$ using dichotomy
**while** $|z_{objective} - z| > 0$ **do**
    $\Delta z \leftarrow z_{objective} - z$
    **while** not IS_IN_BASIN$(z + \Delta z, \text{m})$ **do**
        $\Delta z \leftarrow 0.5 * \Delta z$
    **end while**
    $z \leftarrow z + \Delta z$
    $m \leftarrow$ NEWTON_RAPHSON$(z, \text{Guess} = m)$
**end while**
**return** $m$

---

### 4.1 Initial setup

We first use the classical Newton-Raphson scheme to invert the equation $z = f(m)$ where $z \in \mathbb{C}_+$ is fixed and $f$ is rational. A neat trick which leverages the fact that $f$ is rational and that $z \in \mathbb{C}_+$ is to define:

$$\varphi_z(m) := P(m)/z - Q(z) . \tag{4.1}$$

As such, we have $z = f(m) = \frac{P(m)}{Q(m)} \iff \varphi_z(m) = 0$ . There are several advantages of doing that: (1) Inversion is recast into finding the zero of a polynomial function. (2) Since we have $\lim_{z \to \infty} M(z) = 0$, if $z$ is large in modulus, $m = 0$ is a natural starting point for the algorithm when $z$ is large.

It is well-known that the Newton-Raphson scheme fails unless the initial guess $m_0 \in \mathbb{C}$ belongs to a basin of attraction for the method. And, provided such a guarantee, the Newton-Raphson scheme is exceptionally fast with a quadratic convergence speed. Kantorovich's seminal work in 1948 provides such a guarantee locally. For the reader's convenience, we give in Appendix D the pseudo-code for the Newton-Raphson algorithm (Algorithm 2), as well as a reference for the optimal form of the Kantorovich criterion (Theorem D.1).

Therefore, we assume that we have at our disposal a function $(z, m) \mapsto$ IS_IN_BASIN$(z, m)$ which indicates if the Kantorovich criterion is satisfied for $\varphi_z$ at any $m \in \mathbb{C}$. It is particularly easy to program with $\varphi_z$ polynomial.

### 4.2 Newton lilypads: Doubling strategies and chaining

Now we have all the (local) ingredients in order to describe a global strategy which solves in $m \in \mathbb{C}$ the equation $\varphi_z(m) = 0$ .

First, one has to notice that this problem is part of a family parametrized by $z \in \mathbb{C}_+$. And the solution is $m \approx 0$ for $z$ large. Therefore, one can find a proxy solution for $z \in \mathbb{C}_+$ high enough. This is done thanks to a doubling strategy until a basin of attraction is reached.

Second, if a proxy $(z, m)$ is available, we can use the Newton-Raphson algorithm to find a solution $(z + \Delta z, m + \Delta m)$ starting from $m$. To do so, we need $\Delta z$ small enough. This on the other hand is done by dichotomy.

Tying the pieces together allows to chain the different basins of attraction and leads to Algorithm 1. Notice that in the description of the algorithm, we chose to make implicit the dependence in the function $f$, since it is only passed along as a parameter. Technically, $f$ is a parameter for all three functions NEWTON_RAPHSON, IS_IN_BASIN, NEWTON_LILYPADS.

The discussion leading to this algorithm, combined with the Kantorovich criterion yields:

**Theorem 4.1.** *Given $f : m \mapsto M^{\langle -1 \rangle}(m)$ and $z \in \mathbb{C}^+$, Algorithm 1 has guaranteed convergence. Moreover it returns $m = M(z)$ i.e. the (unique) holomorphic extension of the inverse of $f$ in the neighborhood of $0$.*

# 5  On the experiment

To leverage the numerical scheme, we designed the following experiment whose results are reported on Table 1.1. Consider a classical Multi-Layer Perceptron (MLP) with $L = 4$ layers, feed-forward and fully-connected with ReLU non-linearities. The MLP's architecture is determined by the vector $\lambda = (\lambda_0, \lambda_1, \ldots, \lambda_L)$ while the gains at initialization are determined by the vector $\sigma = (\sigma_1, \sigma_2, \ldots, \sigma_L)$.

By randomly sampling the vector $\lambda$, we explore the space of possible architectures. In other to have balanced architectures, we chose independent $\lambda_i$'s with $\mathbb{E}(\lambda_i) = 1$. Likewise, we also sample different possible gains. Hence we find ourselves with several MLPs architectures each with it's unique initialization. The spectral distributions are computed thanks to our Algorithm 1.

As shown by the parameter count in Fig. 1.2, some architectures are quite large for task at hand. To guarantee convergence despite the changing architectures, we intentionally chose a low learning rate and did 50 epochs of training. For 200 MLPs, the experiment takes 10 hours of compute on a consumer GPU (RTX 2080Ti). This is to be contrasted with less than one minute of CPU compute for the spectral measure $\nu_J$.

To control for any stochastic variability, we also trained multiple instances for each MLP, and the shown results are averages between a few runs. Finally we calculate the correlations between the accuracy on the test set and percentiles of the spectral distribution $\nu_J$. Notice that, a posteriori, the FPT regime is justified because the results are coherent despite various width scales.

# 6  Conclusion

In summary, this paper developed FPT in the rectangular setup which is the main tool for a metamodel for the stability of neural network. Then we gave a method for the numerical resolution both fast and with theoretical guarantees. Finally, confirming the initial claim that stability is a good proxy for performance, we empirically demonstrated that the accuracy of a feed-forward neural network after training is highly correlated to higher quantiles of the *theoretically computed* spectral distribution using this method.

Regarding the importance of hyperbolicity, we surmise that a few large singular values allow the SGD to escape local minima without compromising overall stability.

A challenging problem would be to accommodate for skip-connections. In this case the chain rule used for back-propagation changes in a fundamental way. The Jacobian cannot be approximated by a simple product of free matrices as the same free variable will appear at multiple locations.

# 7  Acknowledgements

R.C. recognizes the support of French ANR STARS held by Guillaume Cébron as well as support from the ANR-3IA Artificial and Natural Intelligence Toulouse Institute.

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
