# A   Appendix: More figures and tables

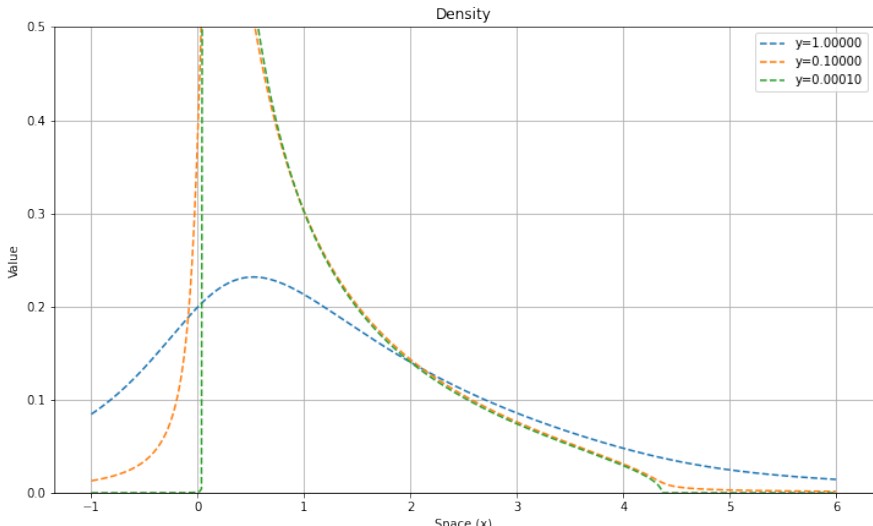

Figure A.1: Plot of the probability density $\frac{-1}{\pi}\Im G_\mu\left(\cdot + iy\right)$ for $y \in \{1, 10^{-1}, 10^{-4}\}$. Here $\mu$ is the multiplicative free convolution of three Marchenko-Pastur distributions, with different parameters.

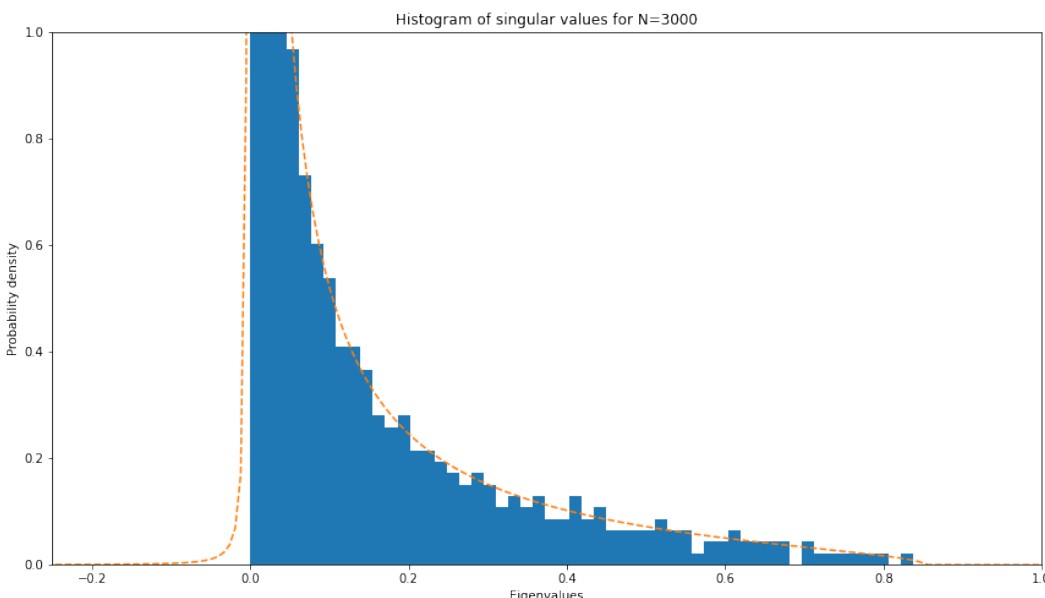

Figure A.2: Plot of density of singular values for the Jacobian matrix $J$. The network has constant width and ReLU non-linearities. Monte-Carlo sampling uses matrices of size $N_\ell = N = 3000$.

# B   Appendix: Table of non-linearities

Let us now collect the various required formulas, and specialize them to a selection of non-linearities: ReLU, Hard Tanh and Hard Sine. These non-linearities are very tractable hence the choice. As discussed in the introduction, the empirical distribution of $D_\ell^{(\mathbf{N})}$ converges to the law of $\phi_l'\left(\sqrt{q^\ell}\mathcal{N}\right)$.

From this observation was deduced in [20] the following formula:

$$M_{D_\ell^2}(z) = \sum_{k \geq 1} \frac{m_k(D_\ell^2)}{z^k} = \mathbb{E}\left[\frac{\phi'_\ell(\sqrt{q^\ell}\mathcal{N})^2}{z - \phi'_\ell(\sqrt{q^\ell}\mathcal{N})^2}\right] , \tag{B.1}$$

$$S_{D_\ell^2}(m) = \frac{1+m}{mM_{D_\ell^2}^{\langle -1 \rangle}(m)} , \tag{B.2}$$

| | Linear | ReLU | Hard Tanh | Hard sine / Triangle | |
|---|---|---|---|---|---|
| $\phi_\ell(h)$ | $h$ | $[h]_+$ | $[h+1]_+ - [h-1]_+ - 1$ | $\frac{2}{\pi}\arcsin \circ \sin(\frac{\pi}{2}h)$ | |
| $M_{D_\ell^2}(z)$ | $\frac{1}{z-1}$ | $\frac{1}{2}\frac{1}{z-1}$ | $C_\ell \frac{1}{z-1}$ | $\frac{1}{z-1}$ | |
| $m_k(D_\ell^2)$ | $1$ | $\frac{1}{2}$ | $C_\ell$ | $1$ | |
| $S_{D_\ell^2}(m)$ | $1$ | $\frac{1+m}{\frac{1}{2}+m}$ | $\frac{1+m}{C_\ell+m}$ | $1$ | |
| $g_\ell(q)$ | $q$ | $\frac{q}{2}$ | $2qC_\ell - \sqrt{\frac{2q}{\pi}}e^{-\frac{1}{2q}}$ | $\frac{1}{3} + \frac{4}{\pi^2}\sum_{n \geq 1}\frac{(-1)^n}{n^2}e^{-q\frac{\pi^2 n^2}{2}}$ | |

Table B.1: Table of formulas for moment generating functions and $S$-transforms.

In Table 1.1, the reader shall find the formulae of $M_{D_\ell^2}$, $S_{D_\ell^2}$ and the recurrence relation given by $f_\ell$ where

$$f_\ell\left(q^{\ell-1}\right) = \sigma_{W_\ell}^2 g_\ell(q^{\ell-1}) + \sigma_{b_\ell}^2 ,$$
$$\text{with } g_\ell(q) := \mathbb{E}\left[\phi_\ell\left(\sqrt{q}\mathcal{N}\right)^2\right] .$$

and

$$C_\ell = \mathbb{P}\left(0 \leq \mathcal{N} \leq \frac{1}{\sqrt{q^\ell}}\right) .$$

*Proof of formulas in Table B.1:* ReLU and Hard Tanh have already been computed in [20].

**Hard Sine:** This is the most tricky formula to establish. If $\phi(x) = \frac{2}{\pi}\arcsin \circ \sin\left(\frac{\pi}{2}x\right)$ and $\widehat{f}(\xi) = \int dx\, f(x)e^{i\xi x}$ is the Fourier transform on $f$, then the application of the Plancherel formula yields:

$$\mathbb{E}\left[\phi\left(\sqrt{q}\mathcal{N}\right)^2\right]$$
$$= \int_{-\infty}^{\infty} dx \frac{e^{-\frac{x^2}{2q}}}{\sqrt{2\pi q}}\phi^2(x)$$
$$= \frac{1}{2\pi}\int_{-\infty}^{\infty} d\xi\, \widehat{\phi^2}(\xi)e^{-q\frac{\xi^2}{2}} .$$

But in term of Fourier series:

$$\phi^2(x) = \sum_{n \in \mathbb{Z}}\left(\widehat{\phi^2}\right)_n e^{i\pi nx}$$

as $\phi^2(x) = x^2$ on $[-1, 1]$ and extended in order to become 2-periodic. In terms of Schwartz distributions:

$$\widehat{\phi^2}(\xi) = \sum_{n \in \mathbb{Z}}\left(\widehat{\phi^2}\right)_n (x \mapsto \widehat{e^{i\pi nx}}) = \sum_{n \in \mathbb{Z}}\left(\widehat{\phi^2}\right)_n 2\pi\delta_{-\pi n}(d\xi) .$$

Hence:

$$\mathbb{E}\left[\phi\left(\sqrt{q}\mathcal{N}\right)^2\right] = \sum_{n \in \mathbb{Z}}\left(\widehat{\phi^2}\right)_n e^{-q\frac{\pi^2 n^2}{2}} .$$

We conclude by computing the Fourier coefficients of $\phi^2$.

$$\left(\widehat{\phi^2}\right)_0 = \frac{1}{2} \int_{-1}^{1} dx \, x^2 = \frac{1}{3} \, .$$

$$\begin{aligned}
\left(\widehat{\phi^2}\right)_n &= \frac{1}{2} \int_{-1}^{1} dx \, x^2 \, e^{-i\pi nx} \\
&= \int_{0}^{1} dx \, x^2 \, \cos(\pi nx) \\
&= -\frac{1}{\pi n} \int_{0}^{1} dx \, 2x \, \sin(\pi nx) \\
&= \frac{2}{(\pi n)^2} \left[x \cos(n\pi x)\right]_0^1 - \frac{2}{(\pi n)^2} \int_{0}^{1} dx \, \cos(\pi nx) \\
&= \frac{2}{(\pi n)^2} (-1)^n \, .
\end{aligned}$$

In the end:

$$\mathbb{E}\left[\phi\left(\sqrt{q}\mathcal{N}\right)^2\right] = \frac{1}{3} + \frac{4}{\pi^2} \sum_{n \geq 1} \frac{(-1)^n}{n^2} e^{-q\frac{\pi^2 n^2}{2}} \, .$$

## C  Appendix: Proofs

### C.1  Proof of Theorem 2.3

For $k \in \mathbb{N}$, we have :

$$\begin{aligned}
Tr\left[\left(\left(A^{(N)}B^{(N)}\right)^T A^{(N)}B^{(N)}\right)^k\right] &= Tr\left[\left(\left(B^{(N)}\right)^T \left(A^{(N)}\right)^T AB\right)^k\right] \\
&= Tr\left[\left(B^{(N)}\left(B^{(N)}\right)^T \left(A^{(N)}\right)^T A\right)^k\right] \, .
\end{aligned}$$

As $\left(A^{(N)}B^{(N)}\right)^T A^{(N)}B^{(N)} \in M_{r_N}(\mathbb{C})$ and $B^{(N)}\left(B^{(N)}\right)^T \left(A^{(N)}\right)^T A^{(N)} \in M_{q_N}(\mathbb{C})$, this shows that

$$M_{\left(A^{(N)}B^{(N)}\right)^T A^{(N)}B^{(N)}}(z) = \frac{q_N}{r_N} M_{B^{(N)}\left(B^{(N)}\right)^T\left(A^{(N)}\right)^T A^{(N)}}(z) \, ,$$

and then

$$M^{\langle -1 \rangle}_{\left(A^{(N)}B^{(N)}\right)^T A^{(N)}B^{(N)}}(m) = M^{\langle -1 \rangle}_{B^{(N)}\left(B^{(N)}\right)^T\left(A^{(N)}\right)^T A^{(N)}}\left(\frac{r_N}{q_N}m\right) \, .$$

Consequently,

$$\begin{aligned}
S_{\left(A^{(N)}B^{(N)}\right)^T A^{(N)}B^{(N)}}(m) &= \frac{1+m}{m M^{\langle -1 \rangle}_{\left(A^{(N)}B^{(N)}\right)^T A^{(N)}B^{(N)}}(m)} \\
&= \frac{1+m}{m M^{\langle -1 \rangle}_{B^{(N)}\left(B^{(N)}\right)^T\left(A^{(N)}\right)^T A^{(N)}}\left(\frac{r_N}{q_N}m\right)} \times \frac{1 + \frac{r_N}{q_N}m}{\frac{r_N}{q_N}m} \times \frac{\frac{r_N}{q_N}m}{1 + \frac{r_N}{q_N}m} \\
&= \frac{r_N}{q_N} \frac{1+m}{1 + \frac{r_N}{q_N}m} S_{B^{(N)}\left(B^{(N)}\right)^T\left(A^{(N)}\right)^T A^{(N)}}\left(\frac{r_N}{q_N}m\right) \, . \quad\quad \text{(C.1)}
\end{aligned}$$

As $\left(A^{(N)}\right)^T A^{(N)}$ and $B^{(N)}\left(^{(N)}\right)^T$ are asymptotically free, taking the limit $N \to +\infty$ and applying Voiculescu's Theorem 2.2 , we get

$$S_{(AB)^* AB}(m) = \alpha \frac{1+m}{1+\alpha m} S_{BB^*}(\alpha m) S_{A^* A}(\alpha m) \, .$$

Moreover, the above equality is true replacing $A$ with the identity $I$. $S_I(m) = 1$ yields:

$$S_{B^*B}(m) = \alpha \frac{1+m}{1+\alpha m} S_{BB^*}(\alpha m) \ .$$

Finally we have

$$S_{(AB)^*AB}(m) = S_{A^*A}(\alpha m)\, S_{B^*B}(m)\ .$$

This concludes the proof.

## C.2   Proof of Theorem 3.1

We assume that $W_\ell$ have i.i.d. entries. Thanks to a swapping trick justified in [17], we can assume that the matrices $D_\ell$ have i.i.d. entries independent from the rest of the network, distributed as $\phi'_\ell(\sqrt{q_\ell}\mathcal{N})$. Notice that we can also replace the $D_\ell$'s by deterministic matrices that use the quantiles of the same distribution. This together with standard results from FPT such as [16] gives asymptotic freeness – see [3] for a more general result reflecting the current state of the art. Therefore, we can apply Theorem 2.3.

Starting from Eq. (1.1) , we get in the infinite width regime and by induction:

$$
\begin{aligned}
&S_{\left(J^{(\mathbf{N})}\right)^T J^{(\mathbf{N})}}(m) \\
&= S_{\left(D_L^{(\mathbf{N})} W_L^{(\mathbf{N})} D_{L-1}^{(\mathbf{N})} W_{L-1}^{(\mathbf{N})} \ldots D_1^{(\mathbf{N})} W_1^{(\mathbf{N})}\right)^T D_L^{(\mathbf{N})} W_L^{(\mathbf{N})} D_{L-1}^{(\mathbf{N})} W_{L-1}^{(\mathbf{N})} \ldots D_1^{(\mathbf{N})} W_1^{(\mathbf{N})}}(m) \\
&= S_{\left(D_L^{(\mathbf{N})} W_L^{(\mathbf{N})} D_{L-1}^{(\mathbf{N})} W_{L-1}^{(\mathbf{N})} \ldots D_2^{(\mathbf{N})} W_2^{(\mathbf{N})} D_1^{(\mathbf{N})}\right)^T D_L^{(\mathbf{N})} W_L^{(\mathbf{N})} D_{L-1}^{(\mathbf{N})} W_{L-1}^{(\mathbf{N})} \ldots D_2^{(\mathbf{N})} W_2^{(\mathbf{N})} D_1^{(\mathbf{N})}}(\lambda_1 m) \\
&\quad \times S_{\left(W_1^{(\mathbf{N})}\right)^T W_1^{(\mathbf{N})}}(m) \\
&= S_{\left(D_L^{(\mathbf{N})} W_L^{(\mathbf{N})} D_{L-1}^{(\mathbf{N})} W_{L-1}^{(\mathbf{N})} \ldots D_2^{(\mathbf{N})} W_2^{(\mathbf{N})}\right)^T D_L^{(\mathbf{N})} W_L^{(\mathbf{N})} D_{L-1}^{(\mathbf{N})} W_{L-1}^{(\mathbf{N})} \ldots D_2^{(\mathbf{N})} W_2^{(\mathbf{N})}}(\lambda_1 m) \\
&\quad \times S_{\left(D_1^{(\mathbf{N})}\right)^2}(\lambda_1 m) S_{\left(W_1^{(\mathbf{N})}\right)^T W_1^{(\mathbf{N})}}(m) \\
&\ \ \vdots \\
&= \prod_{\ell=1}^{L} \left[ S_{\left(D_\ell^{(\mathbf{N})}\right)^2}\left(\prod_{k=1}^{l} \lambda_k m\right) S_{\left(W_\ell^{(\mathbf{N})}\right)^T W_\ell^{(\mathbf{N})}}\left(\prod_{k=1}^{l-1} \lambda_k m\right) \right] \\
&= \prod_{\ell=1}^{L} \left[ S_{\left(D_\ell^{(\mathbf{N})}\right)^2}(\Lambda_\ell m) S_{\left(W_\ell^{(\mathbf{N})}\right)^T W_\ell^{(\mathbf{N})}}(\Lambda_{\ell-1} m) \right] \ ,
\end{aligned}
$$

with the convention $\Lambda_0 = \prod_{k=1}^{0} \lambda_k = 1$.

Under the asumption that the entries of $W_\ell$ are i.i.d., the Marcenko-Pastur Theorem gives

$$S_{\left(W_\ell^{(\mathbf{N})}\right)^T W_\ell^{(\mathbf{N})}}(m) = \frac{1}{\sigma_{W_\ell}^2} \frac{1}{1+\lambda_\ell m},$$

which leads to

$$S_{\left(J^{(\mathbf{N})}\right)^T J^{(\mathbf{N})}}(m) = \prod_{\ell=1}^{L} \left( S_{\left(D_\ell^{(\mathbf{N})}\right)^2}(\Lambda_\ell m) \frac{1}{\sigma_{W_\ell}^2} \frac{1}{1+\Lambda_\ell m} \right) \ .$$

We thus have

$$M^{\langle -1 \rangle}_{\left(J^{(\mathbf{N})}\right)^T J^{(\mathbf{N})}}(m) = \frac{m+1}{m S_{\left(J^{(\mathbf{N})}\right)^T J^{(\mathbf{N})}}(m)}$$

$$= \frac{m+1}{m \prod_{\ell=1}^{L} \left( S_{\left(D_\ell^{(\mathbf{N})}\right)^2}(\Lambda_\ell m) \frac{1}{\sigma_{W_\ell}^2} \frac{1}{1+\Lambda_\ell m} \right)}$$

$$= \frac{(m+1)\prod_{\ell=1}^{L} \sigma_{W_\ell}^2 (1+\Lambda_\ell m)}{m \prod_{\ell=1}^{L} S_{\left(D_\ell^{(\mathbf{N})}\right)^2}(\Lambda_\ell m)}$$

$$= \frac{(m+1)\prod_{\ell=1}^{L} \sigma_{W_\ell}^2 (1+\Lambda_\ell m)}{m \prod_{\ell=1}^{L} S_{\left(D_\ell^{(\mathbf{N})}\right)^2}(\Lambda_\ell m)} \, .$$

# D  Appendix: On the classical Newton-Raphson scheme

---
**Algorithm 2** Newton-Raphson scheme for a rational function $f$

---
**Name:** NEWTON_RAPHSON
**Input:**
Numerical precision: $\varepsilon > 0$ (Default: $10^{-12}$),
Image value: $z \in \mathbb{C}_+$,
Polynomials: $P, Q$ such that $f = \frac{P}{Q}$,
(Optional) Guess: $m_0 \in \mathbb{C}$, (Default: $m_0 = 0$).
**Code:**
$m \leftarrow m_0$
**while** True **do**
   value $\leftarrow \varphi_z(m)$      # See Eq. (4.1)
   **if** $|value| < \varepsilon$ **then**
     **return** $m$
   **end if**
   grad $\leftarrow \varphi_z'(m)$
   $m \leftarrow m - value/grad$
**end while**

---

Here we give the optimal Kantorovich criterion from [10] adapted to this paper. Fix $z \in \mathbb{C}_+$ and recall that $\varphi_z$ in Eq. (4.1) is the map whose zero we want to find.

**Theorem D.1** (Kantorovich's criterion, [13]). *Consider a starting point $m_0 \in \mathbb{C}$, and define:*

$$\delta := \left| \frac{\varphi_z(m_0)}{\varphi_z'(m_0)} \right| \, , \quad \kappa := \left| \frac{1}{\varphi_z'(m_0)} \right| \, .$$

*If the starting point satisfies $h := \delta\kappa\lambda < \frac{1}{2}$, where*

$$\lambda := \sup_{|m-m_0| \leq t^*} |\varphi_z''(m)| \, , \quad t^* := \frac{2\delta}{1+\sqrt{1-h}} < 2\delta \, .$$

*Then, the Newton-Raphson scheme, starting from $m_0$ converges to $m^*$ such that $\varphi_z(m^*) = 0$. Furthermore, the convergence at each step is at least quadratic.*

# E  Appendix: Moments of J

We can reach an early understanding of the behavior of $J$'s singular values by computing mean and variance. For ease of notation, we write:

$$m_k^{(s)}(A) = m_k(A^T A)$$

for any operator $A$, which admits a measure of singular values. We have under the assumptions that the entries of $W_\ell$ are i.i.d. :

$$m_1^{(s)}(J) = \prod_{\ell=1}^{L}\left(m_1^{(s)}(D_\ell)m_1^{(s)}(W_\ell)\right) = \prod_{\ell=1}^{L}(c_\ell\sigma_{W_\ell}^2), \tag{E.1}$$

$$m_2^{(s)}(J) - m_1^{(s)}(J) \tag{E.2}$$
$$= m_1^{(s)}(J)^2\left(\sum_{\ell=1}^{L}\Lambda_\ell\left(\frac{m_2^{(s)}(D_\ell) - m_1^{(s)}(D_\ell)^2}{m_1^{(s)}(D_\ell)^2} + \frac{m_2^{(s)}(W_\ell) - m_1^{(s)}(W_\ell)^2}{m_1^{(s)}(W_\ell)^2}\right)\right)$$
$$= \left(\prod_{\ell=1}^{L}c_\ell^2\sigma_{W_\ell}^4\right)\left(\sum_{\ell=1}^{L}\Lambda_\ell\left(\frac{1-c_\ell}{c_\ell} + \lambda_\ell\right)\right).$$

Under the asumption that $W_\ell^T W_\ell = \sigma_{W_\ell}I_{N_{\ell-1}}$, we find the same $m_1^{(s)}(J)$ and :

$$m_2^{(s)}(J) - (m_1^{(s)}(J))^2 = \sum_{\ell=1}^{L}\left(\Lambda_\ell\left(\frac{m_2^{(s)}(D_\ell)}{m_1^{(s)}(D_\ell)^2} - 2\right)\right)\prod_{\ell=1}^{L}(c_\ell^2\sigma_{W_\ell}^4) \tag{E.3}$$
$$= \sum_{\ell=1}^{L}\left(\Lambda_\ell\left(\frac{1}{c_\ell} - 2\right)\right)\prod_{\ell=1}^{L}(c_\ell^2\sigma_{W_\ell}^4).$$

These formulas need to be interpreted:

- Variance grows with $L$, showing increased instability with depth.

- Larger layers, relative to $N_0$, give larger $\Lambda_\ell$'s and thus the same effect.

*Proof: Computations of moments.* The following remark is useful in the computation of moments.

**Remark E.1** (Moments). *At the neighborhood of $z \sim \infty$:*

$$M_\mu(z) = \frac{m_1(\mu)}{z} + \frac{m_2(\mu)}{z^2} + \mathcal{O}\left(z^{-3}\right).$$

*By inversion, at the neighborhood of $m \sim 0$:*

$$M_\mu^{\langle-1\rangle}(m) = \frac{m_1(\mu)}{m} + \mathcal{O}\left(1\right).$$

$$M_\mu^{\langle-1\rangle}(m) = \frac{m_1(\mu)}{m} + \frac{m_2(\mu)}{m_1(\mu)} + \mathcal{O}\left(m\right).$$

*Hence:*

$$S_\mu(m) = \frac{1+m}{m_1(\mu) + m\frac{m_2(\mu)}{m_1(\mu)} + \mathcal{O}\left(m^2\right)}$$
$$= \frac{1}{m_1(\mu)}(1+m)\left(1 - m\frac{m_2(\mu)}{m_1(\mu)^2} + \mathcal{O}\left(m^2\right)\right)$$
$$= \frac{1}{m_1(\mu)} + \frac{m}{m_1(\mu)}\left(1 - \frac{m_2(\mu)}{m_1(\mu)^2}\right) + \mathcal{O}\left(m^2\right).$$

Thanks to this, we can prove Eq. (E.1) and (E.2). By Remark E.1 and Theorem 3.1, we have as $m \to 0$:

$$S_{J^T J}(m) = \prod_{\ell=1}^{L} \left[ S_{D_\ell^2}(\Lambda_\ell m) \, S_{W_\ell^T W_\ell}(\Lambda_\ell m) \right]$$

$$= \prod_{\ell=1}^{L} \Big[ \left( \frac{1}{m_1^{(s)}(D_\ell)} + m \frac{\Lambda_\ell}{m_1^{(s)}(D_\ell)} \left( 1 - \frac{m_2^{(s)}(D_\ell)}{m_1^{(s)}(D_\ell)^2} \right) + \mathcal{O}\left(m^2\right) \right)$$

$$\left( \frac{1}{m_1^{(s)}(W_\ell)} + m \frac{\Lambda_\ell}{m_1^{(s)}(W_\ell)} \left( 1 - \frac{m_2^{(s)}(W_\ell)}{m_1^{(s)}(W_\ell)^2} \right) + \mathcal{O}\left(m^2\right) \right) \Big]$$

$$= \left[ \prod_{\ell=1}^{L} \frac{1}{m_1^{(s)}(D_\ell) m_1^{(s)}(W_\ell)} \right] \prod_{\ell=1}^{L} \left[ 1 + m\Lambda_\ell \left( 2 - \frac{m_2^{(s)}(D_\ell)}{m_1^{(s)}(D_\ell)^2} - \frac{m_2^{(s)}(W_\ell)}{m_1^{(s)}(W_\ell)^2} \right) + \mathcal{O}\left(m^2\right) \right].$$

Identifying the first order term, one finds indeed Eq. (E.1). Continuing the previous computation:

$$S_{J^T J}(m) = \frac{1}{m_1^{(s)}(J)} + \frac{m}{m_1^{(s)}(J)} \left( \sum_{\ell=1}^{L} \Lambda_\ell \left( 2 - \frac{m_2^{(s)}(D_\ell)}{m_1^{(s)}(D_\ell)^2} - \frac{m_2^{(s)}(W_\ell)}{m_1^{(s)}(W_\ell)^2} \right) \right) + \mathcal{O}\left(m^2\right).$$

Applying Remark E.1 again for $S_{J^T J}$, we get :

$$\sum_{\ell=1}^{L} \Lambda_\ell \left( 2 - \frac{m_2^{(s)}(D_\ell)}{m_1^{(s)}(D_\ell)^2} - \frac{m_2^{(s)}(W_\ell)}{m_1^{(s)}(W_\ell)^2} \right) = 1 - \frac{m_2^{(s)}(J)}{m_1^{(s)}(J)^2}$$

which is equivalent to Eq. (E.2).

We conclude by specializing to classical weight distributions. Under the assumption that the entries of $W_\ell$ are i.i.d. we have $m_1^{(s)}(W_\ell) = \sigma_{W_\ell}^2$ and $m_2^{(s)}(W_\ell) = \sigma_{W_\ell}^4(1 + \lambda_\ell)$ which gives

$$m_2^{(s)}(J) = \left( 1 - \sum_{\ell=1}^{L} \left( \Lambda_\ell \left( 1 - \frac{m_2^{(s)}(D_\ell)}{m_1^{(s)}(D_\ell)^2} - \lambda_\ell \right) \right) \right) \prod_{\ell=1}^{L} \left( m_1^{(s)}(D_\ell)^2 \sigma_{W_\ell}^4 \right)$$

$$m_2^{(s)}(J) - (m_1^{(s)}(J))^2 = \sum_{\ell=1}^{L} \left( \Lambda_\ell \left( \frac{m_2^{(s)}(D_\ell)}{m_1^{(s)}(D_\ell)^2} + \lambda_\ell - 1 \right) \right) \prod_{\ell=1}^{L} \left( m_1^{(s)}(D_\ell)^2 \sigma_{W_\ell}^4 \right).$$

$\square$

# F   Appendix: More details on the benchmarks and the experiment

Recall that we have provided an anonymized Github repository at the address:

`https://github.com/redachhaibi/FreeNN/`

## F.1   On the benchmarks of Fig. 1.1

The figure presents the computational time required for computing the density of $\nu_J$. Let us make the following comments:

- Pennington and al. 's algorithm has been implemented using a native root finding procedure. As such, it is much more optimized than our code.

- A closer examination of the timings shows that Newton lilypads scales sublinearly with the number of required points. This is easily understood by the fact that smaller $N$ requires the computation of more basins of attraction per point.

- Also, the Monte-Carlo method used matrices of size $n = 3000$. Not only Monte-Carlo is imprecise because of the noise, but its performance scales very poorly with $n$ since one needs to diagonalize ever larger matrices. Of course, Monte-Carlo remains the easiest method to implement. But the deterministic methods compute the density up to machine precision.

## F.2 Description of the experiment

Recall that we considered a classical Multi-Layer Perceptron (MLP) with $L = 4$ layers, feed-forward and fully-connected with ReLU non-linearities. The MLP's architecture is determined by the vector $\lambda = (\lambda_0, \lambda_1, \ldots, \lambda_L)$. The initialization follows the Xavier normal initialization [9] as implemented in Pytorch [19]. Thus the full vector of variances of this initialization $\sigma = (\sigma_1, \sigma_2, \ldots, \sigma_L)$ is determined up to a multiplicative factor called the gain..

First, we sample random architectures. We chose

- the $\lambda_i$ to be i.i.d. and uniform on $\{\frac{1}{4}, \frac{1}{3}, \frac{1}{2}, \frac{2}{3}, 1.0, \frac{3}{2}, 2, 3, 4\}$. As such the mean is $\mathbb{E}\lambda_i = 1$ in order to obtain relatively balanced architectures.
- the single gain is taken as a uniform random variable on $\{\frac{1}{4}, \frac{1}{2}, 1, 2, 4\}$.

Secondly, we compute the spectral distributions predicted by FPT for each random architecture. Some architectures are very unbalanced and are discarded.

Thirdly, we train multiple instances of MLPs and record the learning curves.

In the end, by considering FPT quantiles and the final test accuracy for each MLP, we obtain data which amenable to a statistical analysis.

Finally, MNIST and FashionMNIST both have the same format: grayscale images of 28x28 pixels. To be able to train exactly the same MLPs on all three datasets, we applied a grayscale transformation and resizing to 28x28 to CIFAR10.

## F.3 Final remarks

Our sampling procedure does not consider large fluctuations in the $\lambda_i$ 's and focuses on balanced architectures. Likewise, the gains at initialization do not deviate much from the classical [9]. It is important to recognize that without the insight of FPT, the scalings applied to such initializations are already normalized so that spectral measures do converge.

As such, we never encounter truly problematic gradient vanishing or gradient explosion, which completely sabotage the convergence of the neural network. Our refined FPT metrics are arguably "second order corrections". Nevertheless, it is surprising the 90th percentile in Table 1.1 highly correlates to the final test accuracy after training. In the end, tuning FPT metrics does not amount to second order corrections.