# OpenReview forum: "Free Probability for predicting the performance of feed-forward fully connected neural networks"
_NeurIPS.cc/2022/Conference — NeurIPS 2022 Accept_

### Official Review · Reviewer_whcH · 2022-06-17

**Rating:** 6
**Confidence:** 3
**Soundness:** 4 excellent
**Presentation:** 2 fair
**Contribution:** 4 excellent

**Summary:**

In this paper, the authors considered the problem of computing the limiting eigenvalue/singular value distribution of the input-output Jacobian matrix of a fully connected "deep" neural network model having random weights, in the limit where the network widths go to infinity.

The contribution of this paper is threefold:
1. extend the analysis proposed by Pennington et al. in a sequence of papers to the setting of non-square weight matrices by introducing the tool of "rectangular multiplicative free convolution"; and
2. propose a more accurate and faster computational scheme to solve the limiting distribution (see Figure 1), with theoretical guarantees; and
3. apply these theoretical and computational results to empirically show that the above eigenvalue distribution is closely connected to the accuracy of networks, at least in some cases (see Figure 2).



**Questions:**

Some minor issues/remarks:
* The claim "stability is a good proxy for performance" in the introduction should be supported by theoretical, empirical, and/or related previous efforts
* Please provide some references for the term "edge of chaos" in line 44, which may not be familiar to all the readers.
* Please provide some references for the discussions in the paragraph AFTER line 47.
* line 74: I do NOT understand the claim "In the limit, asymptotic freeness will also hold": this is in fact an ASSUMPTION, right? And the notion of "asymptotic freeness" only makes sense asymptotically? Or did I misunderstand something?
* Just a side remark, but I believe it is of interest to mention after Equation (7) whether the operator of multiplicative free convolution is "exchangeable".
* The authors constantly mentioned in the paper "Pennington et al.", but I personally believe it is more clear and more precise to cite the exact paper and the sequence of papers that this work is building upon.
* I wonder if the conclusion drawn from Figure 2 remains valid on more involved ML problems such as CIFAR10/100 and ImageNet image classification.
* line 155: is not associative, i.e., for ...
* line 188: I believe ReLU is more commonly used than ReLu.
* Please provide a few references for the paragraph from line 212 to 217 for better readability.

**Limitations:**

The authors have adequately addressed the technical limitations of their approach.
This work is mainly theoretical and I do not envision any potential negative social impact of this work.

**Strengths And Weaknesses:**

**Strengths**: Despite having some knowledge of the theory of large dimensional random matrices, I am not an expert in free probability theory and it is less evident for me to evaluate the contribution of this work.

The first contribution of this work is of significant practical interest, and the second contribution of this work is, to the best of my knowledge, long believed to be challenging in the field, and the proposed result is worth publishing.

With that said, I strongly recommend that:
1. the Area Chairs and Senior Area Chairs could find an external reviewer for this paper (I can propose a few names if that helps) to better evaluate the (theoretical and computational) contribution of this work; and
2. the authors submit a longer and more detailed version of this paper to a stats/applied math journal since the proposed approaches are of independent interest.

**Weaknesses**: The paper, in its current form, is NOT easily accessible to a large AI/ML audience and should be carefully revised and better organized: for instance, most of the contents in the introduction are in fact "preliminaries", and to introduce the concept of multiplicative free convolution the authors discussed the case of square weight matrices but the scenario considered in this paper is in fact "varying width profiles". In short, the paper can be reorganized to be made clear.

---

> ### Author Response · Authors · 2022-08-02
> **Answers to Reviewer whcH**
>
> Thank you for the quality feedback. Here are the answers to the points stating questions:
>
> - *stability is a good proxy for performance: *
>
> Here by stability we mainly mean the lack of gradient explosions and decay, two issues that are known to hinder training and generalization performances.
> More generally, in numerical analysis, performance, convergence and stability always go hand in hand. We have added a reference to the paper.
>
> - *References for the term "edge of chaos" in line 44:*
>
> Done.
>
> - *Please provide some references for the discussions in the paragraph AFTER line 47.*
>
> We have added a reference to the paper.
>
> - *line 74: I do NOT understand the claim "In the limit, asymptotic freeness will also hold": this is in fact an ASSUMPTION, right? And the notion of "asymptotic freeness" only makes sense asymptotically? Or did I misunderstand something?*
>
> It is not an assumption. It is a consequence of the hypotheses on the randomness and the feed-forward structure. Asymptotic freeness is a step in the proof of Theorem 3.1 using recent results of Sawin-Pastur. See the first paragraph of the proof.
>
> - *I wonder if the conclusion drawn from Figure 2 remains valid on more involved ML problems such as CIFAR10/100 and ImageNet image classification.*
>
> We have extended the paper and provided results on FashionMNIST and CIFAR10: For all datasets, we have obtained very similar correlations: MNIST (0.89%), FashionMNIST (87%), CIFAR10 (83%). These results were added to the paper in  table [1.1].

---

> > ### Comment · Reviewer_whcH · 2022-08-08
> > **Reply to authors' responses**
> >
> > I thank the authors for their detailed responses and acknowledged that I've read the authors' responses and other reviewers' comments. I would consider giving a final rating after further consideration and discussion with other reviewers.

---

### Official Review · Reviewer_GQUV · 2022-07-04

**Rating:** 6
**Confidence:** 3
**Soundness:** 3 good
**Presentation:** 2 fair
**Contribution:** 3 good

**Summary:**

This work extends the free probability theory in the rectangular setup and gives a new method for computing the spectral density of the Jacobian of a general feed-forward neural network with theoretical guarantees. In addition, it also empirically demonstrates that the accuracy of a feed-forward neural network after training is highly correlated to higher quantiles of the theoretically computed spectral distribution.

**Questions:**

- How is the 90th quantile selected? What about other quantiles or other summary statistics? Is a single quantile alone sufficient to quantify the model performance?
- In Line 167, it states that "$N_{l-1}/N_l \rightarrow \lambda_l \ge 0$". Should this be "$\ge$ 0" or "> 0"?

**Limitations:**

The authors addressed the limitation in the final paragraph of the manuscript. Regarding the potential societal impact of this work, I do not see there is any.

**Strengths And Weaknesses:**

Strengths:
- This work is well-motivated: focusing on understand the spectrum of the Jacobian for varying architectures and initializations, providing an effective computation method, and preventing unnecessary tuning of neural networks based on these results.
- On the theoretical side, the authors generalize previous result to rectangular matrices and provide an associate multiplicative convolution operation, which appears novel.
- On the computation side, they present a new method for computing the spectrum leveraging the Newton lilypads, which not only providing theoretical guarantees of always choosing the correct branch but also much faster comparing to existing methods.
- To demonstrate the usefulness of the proposed method, the authors show that the accuracy of a feed-forward neural network after training is highly correlated to higher quantiles of the theoretically computed spectral distribution.

Weaknesses: the main weakness of this paper is the lack of sufficient evidence to support the claims of solving the neural network tuning problem.
- As suggested in the title and the Intro, a main focus of this work is trying to "estimate the performance of an architecture before training", thus providing "a solution to the problem of tuning neural networks". However, there is only indirect evidence that the correlations between the accuracy on the test set and the 90th percentile of the spectral distribution is considerably high for various 4-layer MLPs trained on the MNIST benchmark. This is indeed intriguing, but it is far from a solution to network tuning. The choice of 90th percentile seems arbitrary. Is the 90th quantile the best summary statistics of the spectral density for estimating model performance? What about other quantiles? Or this particular quantile alone is sufficient to quantify the model performance?
- There is lack of discussion of how to apply this metric, 90th percentile of the theoretically computed spectral distribution for example, in practice. Also, how to avoid the vanishing and explosive regimes? It would also be nicer to provide more results on multiple benchmarks.
- In Section 1.1, the authors argued that "smaller networks with more spread-out $\nu_J$ perform better than larger counterparts". But I could not find any supporting evidence for this. In addition, the quantitative results are only concerned with the 90th quantile with little discussion/summarization about the spread-ness of the whole spectral distribution.
- There is existing work discussing the relationship between trainability and generalization, [1] for example. It argues that "trainability and generalization are distinct notions that are, at least in this case, at odds with one another. Indeed, good conditioning of the NTK (which is a necessary condition for training) seems necessarily to lead to poor generalization performance. It will be interesting to see whether these results carry over in shallower and narrower networks." The last question seems can be answered from this work. It would be interesting if the authors could discuss more on this.

Typo:
- In Line 186, it should be $M_{J^TJ}$ instead of $M_J$.

[1] L. Xiao, J. Pennington, and S. Schoenholz. Disentangling trainability and generalization in deep neural networks. ICML 2020.

---

> ### Author Response · Authors · 2022-08-02
> **Answers to Reviewer GQUV**
>
> We thank Referee GQUV for the quality feedback. We agree that providing limited empirical benchmarks was the main weakness of the paper. We have addressed it by including FashionMNIST and CIFAR10 and by considering all quantiles instead of the 90th. The new results are now reported in Table 1.1 and are in agreement with MNIST. This will hopefully allow the Referee to raise his rating of our paper.
>
> Comments on other weaknesses:
> - “far from a solution to network tuning”: Our solution is certainly not general enough to handle all networks. The problem of architecture tuning is very hard. This is however to be contrasted with the extremely low computational cost and mathematical soundness of the approach. We hope that our work will be a stepping stone for future research. Here, we have only provided a method to assess fully connected architectures with rectangular matrices.
> - Choice of the 90th percentile: We have added results to the paper including all percentiles in Table 1.1 and A.1. When all are considered we consistently obtain higher correlations MNIST (84 to 89%), FashionMNIST(81 to 87%), CIFAR10 (75 to 83%). The table A.1 clearly shows that when taken individually the 90th percentile is the most correlated with test performances -- hence the choice.
> - “smaller networks with more spread-out distribution **can** perform better than larger counterparts”: This sentence has been rephrased in the paper. The explanation is as follows. In the scatter plots of accuracy vs Log10 of 90th quantile (Fig A.3), everytime a lower color appears north-east of a higher color, it corresponds to a smaller network outperforming a bigger network.
> - “Relationship between trainability and generalization”: That is a great and deep question! Trainability and generalization seem to be two distinct notions indeed.  They are reported as at odds with each other in the NTK regime.
> While the NTK is a beautiful mathematical object, we do not believe we are in that regime. We start with a theoretical framework based on analyzing stability and hence trainability. However our empirical results are reported on test sets. They thus are more about generalization than trainability. The existence of empirical correlation between trainability indicators (FPT metrics) and generalization indicators (test accuracy) is an argument excluding the NTK regime.
>
> Typo: Well spotted!
>
> Answers to questions:
> - 90th quantile: Answered above
> - “ $\lambda_\ell \geq 0$ or $>0$”: The case =0 is an edge case but it works fine theoretically. We corrected it in order to be on the safe side.

---

> > ### Comment · Reviewer_GQUV · 2022-08-05
> > **Thanks for the response**
> >
> > Thanks for the detailed response. I do believe this work makes nontrivial theoretical and computational extensions of existing work. That's why I am leaning towards acceptance.
> >
> > My biggest concern and the reason of the score 5, as I stated in the original review, is the mismatch between the title/intro and the content of the paper. There is insufficient evidence to support the "solution to the problem of tuning neural networks" claim. This is not about generalness of the solution to handle all networks. A solution only for fully connected networks will be sufficient as long as it works in certain way. Unfortunately, there is no formal description of the "solution" itself: how to use the quantiles of the spectral density to find the adequate model architecture and meanwhile avoid the vanishing and explosive regimes. It would also be somewhat necessary to compare with existing model performance inference methods with limited training or no training at all. On the other hand, it would be different if the empirical observation is considered as the preliminary insight for the theoretical and computational analysis in this work.
> >
> > Z. Zhang, and Z. Jia. GradSign: model performance inference with theoretical insights. ICLR 2022.

---

> > > ### Author Response · Authors · 2022-08-08
> > > **Answer**
> > >
> > > Dear Reviewer GQUV,
> > >
> > > Thank you for the GradSign reference.
> > >
> > > Upon carefully reading that paper, we could perhaps surmise that this is the sort of paper you expected -- given our title and abstract.
> > > We are sorry if we encouraged the confusion, but it was never our aim to write a paper as strongly focused on architecture search.
> > > We aim for a more general and less specialized audience, which is interested in the spectra of Jacobians in relation to stability and performance.
> > > For example, it is easy to imagine tuning learning rates from the knowledge of Jacobians, or automated initialization schemes like Glorot's.
> > > Moreover, FPT metrics require time to be introduced. And the computational aspects are known to be difficult, hence an extensive part of the paper developing that.
> > >
> > > Answers:
> > > - "Mismatch between title/intro and the content of the paper:" In hindsight, we agree. We believe this can be easily fixed though. How about a straight-to-the-point title "Free Probability for predicting the performance of feed-forward neural networks"?
> > > We will change the abstract accordingly.
> > >
> > > - "Formal description of the solution": One needs to follow our experiment. For a set of candidate architectures, compute spectral measures $\nu_J$ thanks to our algorithm. Then:
> > > 1. Discard if vanishing regime which is to say if $\nu_J( [0, 10^{-1}] ) > 0.5$
> > > 2. Discard if explosive regime which is to say if $\nu_J( [10, \infty) ) > 0.5$
> > > 3. Prioritize architectures with larger 90-th quantile i.e.
> > > $ \alpha = \inf \{ ( x \geq 0 \ | \ \nu_J( [0,x] ) \geq 0.9  ) \}$
> > >
> > > Kind regards
> > >
> > > The authors

---

> > > > ### Comment · Reviewer_GQUV · 2022-08-09
> > > > **Thank you**
> > > >
> > > > Thanks for the response! I myself do not think the main theme of this work is focusing on architecture search, but the title/abstract is somewhat misleading. I appreciate the authors' effort to modify the title/abstract to make it more consistent with the main content. Thank you!

---

### Official Review · Reviewer_Qq6Q · 2022-07-09

**Rating:** 7
**Confidence:** 3
**Soundness:** 3 good
**Presentation:** 3 good
**Contribution:** 3 good

**Summary:**

This paper deals with the multiplicative case of the free rectangular convolution and its application to neural networks.

The multiplicative case of the free rectangular convolution, contrarily to the additive one, comes down, as it is noticed i Section 2 of the paper, to the square case (ie to the case of square matrices). The numerical computation algorithm the authors give was a missing pice in this pipeline and it is valuable.

I thus recommend acceptation.

**Questions:**

.

**Limitations:**

.

**Strengths And Weaknesses:**

This paper deals with the multiplicative case of the free rectangular convolution and its application to neural networks.

The multiplicative case of the free rectangular convolution, contrarily to the additive one, comes down, as it is noticed i Section 2 of the paper, to the square case (ie to the case of square matrices). The numerical computation algorithm the authors give was a missing pice in this pipeline and it is valuable.

I thus recommend acceptation.

---

> ### Author Response · Authors · 2022-08-02
> **Answers to Reviewer Qq6Q**
>
> We thank Reviewer Qq6Q for the positive feedback.

---

### Official Review · Reviewer_1DWD · 2022-07-11

**Rating:** 6
**Confidence:** 2
**Soundness:** 3 good
**Presentation:** 1 poor
**Contribution:** 2 fair

**Summary:**

The paper presents an extension of the work by Pennington et al. 2018 for the spectrum of the Jacobian in deep fully connected neural networks at initialization. The result presented in the paper requires the numerical computation of the S-transform and the authors presented a method that out-performs the proposals from Pennington et al. 2018.

**Questions:**

Please see the section "Strengths And Weaknesses"

**Limitations:**

The limitations have not been addressed. I commented on this point in the section "Strengths And Weaknesses".

**Strengths And Weaknesses:**

Strengths:
* The paper presents an improvement over the results of Pennington et al. 2018 to more realistic settings.

Weaknesses:
* The improvement is slight. Although I agree with the authors that the settings of Pennington et al. are far from the DNN used in applications, the same critique applies to their setup: they only consider dense layers.
* I couldn't understand whether they found situations where their generalisation leads to architectures that outperform the one identifiable using Pennington's result.
* The section on the empirical results shows tests on the MNIST dataset. However, on this dataset even a single-layer network can achieve more than 90% accuracy, I honestly find that section not significant unless they show results on simple but more complex datasets (for instance fashion MNIST).
* Overall I found the paper hard to read because it follows a rather unconventional structure. I would suggest shuffling and re-adapting some subsections of section 1 in order to be closer to other papers in ML conferences. For instance, without the intension of forcing this order in particular, the following order seems more natural to me:
   Introduction ->  Randomness -> Modelling -> Contributions -> Structure of the paper -> (maybe this part could go in section 3) Framework -> intuition (this doesn't seem necessary)
* It would be good to add a "related work" section. The work from Pennington is from 2018, I believe progress has been made since then and probably they should be mentioned.
* Connected with the assumption made in section 3 on the equivalence to Gaussian random variables, I would point out the work below in supports "Sebastian Goldt, Bruno Loureiro, Galen Reeves, Florent Krzakala, Marc Mezard, Lenka Zdeborova Proceedings of the 2nd Mathematical and Scientific Machine Learning Conference, PMLR 145:426-471, 2022."

---

> ### Author Response · Authors · 2022-08-02
> **Answers to Reviewer 1DWD**
>
> We thank Reviewer 1DWD for the quality feedback. We hope that the following answers will allow you to raise your rating of the paper. We feel there have been some honest misunderstandings. Pennington et al. do not consider the question of architecture search. Their paper is pioneering in the sense that they were the first in ML to use FPT to describe universal behavior of spectra in NNs. The results of our paper are only comparable to theirs regarding the computational aspects and not regarding the experiment on real NNs. In that regard our improvements regarding the works of Pennington et al. are the following: (1) Our method always works with theoretical guarantees. (2) We show that our algorithm is more than 10x faster.
>
> Other comments:
> - Only MNIST in the experiment: We agree completely. We added FashionMNIST and CIFAR10 . Results on these datasets are very similar to those obtained on MNIST with high correlations: MNIST(89%), FashionMNIST(87%), CIFAR10(83%) and are reported in Table 1.1. These results suggest that our framework has a strong degree of independence towards datasets, as implied by the theory.
>
> - Gaussian equivalence: Thank you for the reference.
> More generally, the Gaussian equivalence is ubiquitous in probability theory and is referred to as "Universality". And the typical example is the Central Limit Theorem which is universal in the original distribution.
> The paper suggested by the Reviewer, is concerned with the universal behavior of SGD with respect to the data distribution.
> So far, our paper was more concerned with the universality in the entries of the weights of NNs and we ignored inputs.  In the context of FPT or Random Matrices, the question of universality is to prove that spectral properties are universal in the distribution of the matrix entries. This is in the literature and our proofs give the necessary references.
> In the updated version of our paper, we notice the universality of behavior for the datasets MNIST, FashionMNIST and CIFAR10.

---

> > ### Comment · Reviewer_1DWD · 2022-08-08
> > **Thank you for reply - clarifications**
> >
> > Dear authors, thank you for the effort in the reply. I am happy with the new simulations that make the result more compelling.
> >
> > I would like the authors to clarify further the first point of their reply. I see that I should have kept more into consideration the computational aspect in the comparison with Pennington et al. 2018, however, I think that we can as well draw a comparison in the terms of the results. The authors are proposing an analysis that allows for arbitrary width of the layers of the network. I wonder what are the consequences of these additional degrees of freedom. For instance, how is the order-chaos transition affected by the different width ratios?

---

> > > ### Author Response · Authors · 2022-08-08
> > > **Discussing the order-chaos transition**
> > >
> > > Dear Reviewer 1DWD,
> > >
> > > Indeed, one can focus on the comparison with Pennington's work, and more specifically the analysis of the order-chaos transition.
> > > In answering the question "how is the order-chaos transition affected by the different width ratios", the key element in this comparison is the depth L.
> > >
> > > - In very deep networks, the order-chaos transition is very sharp.
> > >
> > > A series of papers [1], [2], [3] highlight the importance of dynamical isometry for very deep NNs ($L \gg 100$). (Dynamical isometry = singular values at 1).
> > >
> > > The intuitive explanation is that deep homogenous architectures behave like homogenous dynamical systems. The long-time behavior of dynamical systems (large L for NNs) sharply transitions from order to chaos depending on Lyapunov exponents (Singular values of a slice in NNs). Probably that one could prove the existence of a free convolution semi-group.
> > >
> > > In order to maintain dynamical isometry, Pennington advocates for orthogonal initializations, and certain non-linearities only, which do not give spread-out distributions around 1.
> > >
> > > In such a setting, varying widths would be catastrophic for feed-forward NNs. Among other effects, this creates rank-deficient matrices and therefore singular values at 0. The long products exacerbate the problem further.
> > >
> > > - Our message is that not so deep networks, the transition is not sharp at all.
> > >
> > > Rank deficient jacobians due to big variations in width are not that problematic.
> > > Even more shockingly to us, somewhat spread-out distributions are desirable.
> > >
> > > In our opinion, this nuance about the edge of chaos was the most interesting story to tell.
> > > And our experiment was aimed at showing that: lower quantiles are not that important, and higher quantiles matter.
> > > The very large depth is peculiar as it forces everything to collapse to 0 or infty.
> > >
> > > In short, for smaller L, the order-chaos transition becomes not sharp, so that extra degrees of freedom like width can come to play.
> > > For large L, one cannot play with widths while maintaining dynamical isometry.
> > >
> > > Additional note 1: Pennington's algorithm in [1] treats the inversion of S transforms as if the desired root for L layers is close to that for L-1.
> > > This is correct in the dynamical isometry regime as the singular values of an operator $A( Id + \delta )$, for small $\delta$, are definitely close to those of $A$.
> > > But this is simply false outside of that regime.
> > >
> > > Additional note 2: As the transition is not sharp, the extra degrees of freedom do not have a simple effect on stability. Their effect is basically understanding the map $(\lambda_1, \dots, \lambda_L) \mapsto \nu_J$, which needs all the technology of FPT. For large L>>100, with dynamical isometry, this map becomes simple and has been described in [1].
> > >
> > > Kind regards
> > >
> > > The authors
> > >
> > > [1] The emergence of spectral universality in deep networks. AISTATS 2018
> > >
> > > [2] Dynamical Isometry and a Mean Field Theory of RNNs: Gating Enables Signal Propagation in Recurrent Neural Networks. ICML 2018
> > >
> > > [3] Dynamical Isometry and a Mean Field Theory of CNNs: How to Train 10, 000-Layer Vanilla Convolutional Neural Networks. ICML 2018

---

### Official Review · Reviewer_PUn2 · 2022-07-12

**Rating:** 7
**Confidence:** 4
**Soundness:** 4 excellent
**Presentation:** 4 excellent
**Contribution:** 3 good

**Summary:**

UPDATED SCORE AFTER REBUTTAL.

The pioneering work of Pennington et al. used free probability to predict the spectrum of the input-output Jacobian. The conditioning of this matrix is known to be important for training stability. The authors generalize those results to sequences of width ratios that need not be identical. Despite knowing the self-consistent equation for the spectrum, obtaining numerical predictions are known to be numerically challenging. The authors propose a new method, which is guaranteed to converge, to numerically solve for the spectrum. Experiments are also performed to show that the 90th percentile of the spectrum is highly correlated with the loss of a fully connected neural network trained on MNIST.

**Questions:**

- Why is there a line of points in Figure 2 at $x\approx-1.6$?
- Correlation for a random sample of architectures is interesting, however, there is still a question of whether this metric could be used iteratively in some kind of hyperparameter search.
- There have been other works using free probability to study ML. Can this algorithm be used for the equations in any of those cases?
- Were other functionals of the spectrum than the 90th percentile considered?

**Limitations:**

No concerns about negative societal impact.



**Strengths And Weaknesses:**

Strengths
- To my knowledge, the lilypad algorithm is an original contribution even in the free probability community and not merely free probability applied to ML. The speed-ups provided over the algorithm of Pennington et al. are significant. Combined with the convergence guarantee the lilypad algorithm appears preferable.
- The paper is clear and easy to read. In particular, the overview of free probability is nice and an accessible introduction.
- It's great to see open sourced code released with the paper.

Weaknesses
- The paper's significance is hurt by only considering the MNIST dataset in the experiments. It should be easy to try CIFAR10 for example to check whether the correlation still holds. This is important given how easy MNIST is as a benchmark.
- Similarly only FC architectures are considered. Even the activation function is unchanged.
- The main contributions are interesting mathematically but they might not be of great interest to the NeurIPS community, given the limitations above in the more practical experiments.

---

> ### Author Response · Authors · 2022-08-02
> **Answers to Reviewer PUn2**
>
> We thank Reviewer PUn2 for the quality feedback. We agree that using only MNIST for experiments was a weakness. We have therefore added FashionMNIST and CIFAR10. We hope this will allow you to raise the rating of our paper.
> For all datasets, we obtain very similar correlations: MNIST (89%), FashionMNIST (87%), CIFAR10 (83%). These results were added to the paper in table [1.1]. This experiment also suggests that our approach is dataset independent, as expected from the theory: at the theoretical level, the FPT metamodel is already dataset independent.
> As such experience and theory agree regarding this aspect.
>
> Comments on other weaknesses:
> - FC architectures: FC layers are inherited from the framework of FPT which requires delocalized entries in order for spectra to concentrate.
> As mentioned in conclusion, convolution layers are simply out of the scope of the theory. They are too structured and thus require a completely separate treatment – and more advances in FPT both at the theoretical and computational level.
> - Other activation functions: We agree that a study of different nonlinearities would be insightful. However in the interest of space and clarity, we have elected to focus on the most used activation (ReLU).
>
> Answers to questions
> - Line at x=-1.6: The exact value is x=-1.62887 and is in log-scale. It corresponds to distributions whose 90th quantile is approx 10^{-1.6} \approx 0.0235.
> This corresponds to the first non-zero point in our grid when computing spectral densities. As such, the line at x=-1.6 is caused by the finite precision of our computations which cannot distinguish between a quantile at 10^{-2} and 10^{-3}.
> - Use in hyperparameter search: The naive approach would simply follow our experiments . One could sample architectures and select those with a spread-out spectral distribution  (high 90th quantile), non-explosive (significant mass < 2) and non-vanishing (significant mass > 0.5).
> Fancier techniques could everage the exploitation-exploration paradigm, using optimal control or reinforcement learning. The latter is definitely worthy of a paper of its own.
> - Can this algorithm be used in other cases: Yes. Operator-valued FPT would be useful to extend the feed-forward setup. Recently it was used in “Adlam, Pennington. The Neural Tangent Kernel in High Dimensions: Triple Descent and a Multi-Scale Theory of Generalization. ICML2020.” to analyze the double descent phenomenon. Chaining Newton-Raphson basins works also for matrix variables and therefore, our technique applies.
> - Other functionals than the 90th quantile: We updated the paper and have added the R factor of a multivariate regression against all quantiles 10, 20, … , 90% to table [1.1]. Considering all quantiles raise correlation levels for all datasets MNIST(84 to 89%), FashionMNIST(81 to 87%), CIFAR10(75 to 83%). We also added an experiment showing that taken individually, the 90th quantile is the most informative (Appendix A. Table A.1.).

---

> > ### Comment · Reviewer_PUn2 · 2022-08-08
> > **Response to authors**
> >
> > I'm grateful to the authors for their response to my review. I think the additional experiments on other datasets certainly strengthen the results, and I will therefore increase my score to accept.
> >
> > One additional comment on the possibility of using FPT for hyperparameter search: Currently the experiments consider correlations in a random sample of architectures. Obviously if some kind of iterative search is performed, this would change the distribution over architectures, and so might reduce the correlation between the metric and performance. Although iterative hyperparameter search is not a direct goal of the paper, I think it's important to acknowledge this potential limitation and point out it requires further study.

---

> > > ### Author Response · Authors · 2022-08-09
> > > **Thanks**
> > >
> > > Dear Reviewer PUn2,
> > >
> > > thank you for your time.
> > >
> > > Regarding your additional comment, we agree. Iterative hyperparameter search indeed brings the challenges you mention:
> > > exploiting the information gained from earlier iterations can strongly bias the exploration scheme.
> > >
> > > Kind regards
> > >
> > > The authors

---

### Meta-Review · Area_Chair_F4ej · 2022-08-20

**Recommendation:** Accept
**Confidence:** Certain

**Metareview:**

The main contribution of this paper is an algorithm to compute the spectral density of the free multiplicative convolution which appears in the expression of the Jacobian of neural networks. Some concerns have been raised regarding the impact on a general ML audience (reviewers PUn2 and 1DWD), and about the presentation of the material (especially abstract and introduction, see review GQUV). I share part of these concerns. At the same time, the proposed “Newton lilypads” algorithm not only provides a speed-up over prior work by Pennington et al., but it appears to be an original and interesting contribution in free probability beyond its ML application. Therefore, I ultimately agree with the reviewers, who have reached a consensus of accepting this paper. The novelty aspect of this paper will make it an interesting addition to the NeurIPS 2022 technical program. As a final note, I would like to strongly encourage the authors to include in the camera ready the additional experiments on FashionMNIST and CIFAR10, as well as the discussions related to the feedback from the reviewers (including, if possible, the suggested title/abstract change).


**Award:**

No

---

### Decision · Program_Chairs · 2022-09-14

Accept